

# Diurnal, weekly, seasonal and spatial variabilities in carbon dioxide flux in different urban landscapes in Sakai, Japan

Masahito UEYAMA[1] and Tomoya ANDO[1]

[1]*Graduate School of Life and Environmental Sciences, Osaka Prefecture University, 1-1, Gakuen-cho,*

*Naka-ku, Sakai, Osaka, 599-8531, Japan*

Corresponding author: M. Ueyama, Graduate School of Life and Environmental Sciences,

Osaka Prefecture University, Sakai, Japan (miyabi-flux@muh.biglobe.ne.jp)

Running title: $CO_2$ fluxes over urban areas in Sakai

Keywords: $CO_2$ emission, eddy covariance, temporal and spatial variations, green fraction

**Abstract**

To evaluate $CO_2$ emissions in urban areas and their temporal and spatial variability, continuous measurements of $CO_2$ fluxes were conducted using the eddy covariance method at three locations in Sakai, Osaka, Japan. Based on the flux footprint at the measurement sites, $CO_2$ fluxes from the three sites were partitioned into five datasets representing a dense urban center, a moderately urban area, a suburb, an urban park, and a rural area. A distinct biological uptake of $CO_2$ was observed in the suburb, urban park, and rural areas in the daytime, whereas high emissions were observed in the dense and moderate urban areas in the daytime. Weekday $CO_2$ emissions in the dense urban center and suburban area were approximately 50% greater than emissions during weekends and holidays, but the other landscapes did not exhibit a clear weekly cycle. Seasonal variations in the urban park, rural area, and suburban area were influenced by photosynthetic uptake, exhibiting the lowest daily emissions or even uptake during the summer months. In contrast, the dense and moderately urban areas emitted $CO_2$ in all seasons. $CO_2$ emissions in the urban areas were high in the winter and summer months, and they significantly increased with the increase in air temperature in the summer and the decrease in air temperature in the winter. Irrespective of the land cover type, all urban landscapes measured in this study acted as net annual $CO_2$ sources, with emissions ranging from 0.5 to 4.9 kg C m$^{-2}$ yr$^{-1}$. The magnitude of the annual $CO_2$ emissions was negatively correlated with the green fraction; areas with a smaller green fraction had higher annual $CO_2$ emissions. Upscaled flux estimates based on the green fraction indicated that the emissions for the entire city were 3.3 kg C m$^{-2}$ yr$^{-1}$, which is equivalent to 0.5 Tg C yr$^{-1}$ or 1.8 Mt $CO_2$ yr$^{-1}$, based on the area of the city (149.81 km$^2$). A network of eddy covariance measurements is useful for characterizing the spatial and temporal variations in net $CO_2$ fluxes from urban areas. Multiple methods would be required to evaluate the rationale behind the fluxes and overcome the limitations in the future.

## 1. Introduction

Cities emit a considerable amount of carbon dioxide ($CO_2$) that is associated with human activities into the atmosphere (Canadell et al., 2007). Urban areas account for only a small percentage of the earth's land surface but emit 30–50% of total anthropogenic $CO_2$ (Mills, 2007; Stterthwaite, 2008), and thus, cities are important sources of the global $CO_2$ emissions. The $CO_2$ emissions among global cities are highly heterogeneous (Mills, 2007; Nordbo et al., 2012), and the temporal variability is high (Velasco and Roth, 2010). To evaluate the spatio-temporal variabilities in $CO_2$ emissions for global cities, studies using multiple methods, such as measurements (Velasco and Roth, 2010) and emission inventories (Oda and Maksyutov, 2011), are currently conducted.

Global $CO_2$ emissions have often been estimated using emission inventories based on point source databases, statistics for national and regional $CO_2$ emissions, and satellite remote sensing (Oda and Maksyutov, 2011). The major challenge for estimating global $CO_2$ emissions is to understand the spatio-temporal dynamics of $CO_2$ emissions in various cities. Because emissions data are used in top-down estimates of the global $CO_2$ budget (Peters et al., 2007; Schmel et al., 2001), a better estimate of $CO_2$ emissions from cities will improve our understanding of the global carbon cycle, including terrestrial and ocean fluxes.

To evaluate $CO_2$ emissions in cities and their temporal and spatial variabilities, continuous measurements of $CO_2$ fluxes have been conducted using the eddy covariance method in various urban landscapes, including dense urban built-up areas (Gioli et al., 2012; Grimmond et al., 2002, 2004; Kotthaus and Grimmond, 2012; Nimitz et al., 2002; Pawlak et al., 2011; Velasco et al., 2005), suburban areas (Bergeron and Strachan, 2011; Coutts et al., 2007; Crawford et al., 2011; Hirano et al., 2015; Moriwaki et al., 2006; Ward et al., 2013), urban parks (Kordowski and Kuttler, 2010), and urban forests (Awal et al., 2010), in several cities. These results have indicated that cities emits a considerable amount of $CO_2$ into the atmosphere from human activities, such as vehicle traffic and household heating in the wintertime. Even in urban parks, $CO_2$ was emitted to the atmosphere due to human activities (Kordowski and Kuttler, 2010). The magnitude of $CO_2$ emissions and its temporal variability depended on the city, associated with the type

of human activities under different climate conditions (Järvi et al., 2012; Moriwaki et al., 2006; Velasco et al., 2016; Ward et al., 2013, 2015), and the role of urban vegetation (Awal et al., 2010; Kordowski and Kuttler, 2010; Peters and McFadeen, 2012; Ward et al., 2015), showing considerable heterogeneities.

Multi-site eddy covariance towers were used to synthesize the data and showed that green fraction was the index that explained the spatial variability in annual $CO_2$ emissions (Nordbo et al., 2012; Velasco and Roth, 2010; Ward et al., 2015), because the green fraction has many possible factors that determine $CO_2$ emissions: a greater green fraction correlates to lesser road and population densities (Nordbo et al., 2012). Upscaling using the green fraction can provide a high-resolution map of direct $CO_2$ emissions from cities. Previous studies have examined the relationship between annual $CO_2$ emissions and the green fraction at a global scale (Nordbo et al., 2012; Velasco and Roth, 2010; Ward et al., 2015). It is unclear whether upscaling $CO_2$ emissions is possible within a city, because multi-site eddy covariance measurements within a city are often unavailable.

In this study, we present diurnal, weekly, seasonal, and spatial variabilities in the $CO_2$ fluxes continuously measured at three different locations within 5 km of each other in Sakai, Osaka, Japan. Considering flux footprint, the data represent five urban landscapes, including a dense urban center, a moderately urban area, a suburb, a rural area, and an urban park. Regardless of the landscape type, all landscapes emitted considerable $CO_2$ annually with different temporal metabolisms, providing a useful overview of anthropogenic $CO_2$ emissions.

## 2. Materials and methods

*2-1. Study sites*

Sakai is the second largest city in Osaka Prefecture, located in western Japan. The population was approximately 842,000 in 2015. Because the city is located on the eastern shore of Osaka Bay, sea breeze circulation is evident throughout the year, except when seasonal winds are not strong. The area is on a uniformly flat plane; the north-south and the east-west slopes are 0.0030° and 0.0024°, respectively. The climate of Sakai is temperate; the mean annual air temperature is 15.9°C, the maximum monthly mean air

temperature was 28.0°C in August, the minimum monthly mean air temperature was
5.2°C in January, and the mean annual precipitation was 1187 mm yr$^{-1}$ between 1981 and
2010 according to the Japanese Meteorological Agency.

The Sakai city center (SAC) site (Fig. 1; Table 1) is located on a tower at the top of a

city office building (34°34′25″N, 135°28′59″E). The population density around the city
center is approximately 12150 km$^{-2}$, based on the Japanese Government Statistics. The
area is a densely built-up urban area with a mean building height of 10.7 ± 3.1 m. Because
the distributions of building heights were highly skewed toward low-height buildings, the
mean building height greater than 20 m was 36 m, which occupied 33% of the total
building area. Many arterial roads and two highways with heavy traffic are present within
the flux footprint. Because industrial and commercial areas are located in the western and
northern parts of the city, those areas are expected to show higher rates of human activity
than locations where residential areas are dominant.

The Oizumi Ryokuchi urban park (IZM) site (Fig. 1; Table 1) is located at the northern

end of the city (34°33′48″N, 135°32′1″E), and was established in 1972. The
measurements were conducted at a tower located at the eastern edge of the park. Because
of the consistent presence of a sea breeze, the tower is mostly located downwind of the
park during the daytime. The land cover of the park consists of 51% trees, 15% grassland,
and 34% other, such as ponds, buildings, pavement, and bare ground. No vehicle traffic
was allowed in the park except parking. Measurements using a plant canopy analyzer
(LAI-2000, LI-COR, Lincoln, Nebraska, USA) showed that the leaf area index of trees
ranged from 3.2 to 5.7 m$^2$ m$^{-2}$ with a mean of 4.3 m$^2$ m$^{-2}$ in the summer months. The mean
and maximum tree heights were estimated as 12.3 ± 4.1 m and 21 m, respectively, using
a digital surface model by Google Earth. The area surrounding the IZM is a mixed
landscape of residential areas and agricultural fields and is characterized as a rural area
(Table 1). The population density of surrounding residences surrounding the IZM is
approximately 7940 km$^{-2}$, based on the Japanese Government Statistics.

The Osaka Prefecture University (OPU) site (Fig. 1; Table 1) is located at the western

edge of Osaka Prefecture University (34°32′50″N, 135°30′10″E). Because the
measurements were conducted on the roof of a building, the flux footprint represents only

a small suburban area. The western part of the site contains a protected forest on a kofun (the ancient burial mound), Mozu Kofungun. The area is characterized as a suburb, consisting of a university, a residential area, small streets, a graveyard, and trees. The mean and maximum tree heights were $13.1 \pm 2.9$ m and 19 m, respectively, and the mean and maximum building heights are $9.1 \pm 2.9$ m and 15 m, respectively.

*2-2. Observations*

We measured $CO_2$ fluxes using the eddy covariance method at the three sites. For SAC, a sonic anemometer (SAT550, Sonic Corp., Tokyo, Japan) and an open-path infrared gas analyzer (LI-7500, LI-COR) were installed on a 16-m tower located at the top of the city office building (111 m above the ground) at the end of November, 2009. For IZM, a sonic anemometer (CSAT3, Campbell Scientific Inc., Logan, Utah, USA) and an open-path infrared gas analyzer (EC150, Campbell Scientific Inc.) were installed 30 m above the ground on a tower at the end of January 2015. For OPU, sonic anemometers and several infrared gas analyzers were installed on a 2-m mast above the rooftop at the edge of the building (16.2 m above the ground) in November 2014. Turbulent fluctuations were recorded at 10Hz using a datalogger (8421, Hioki, Japan) for SAC and dataloggers (CR1000, Campbell Scientific Inc.) for IZM and OPU.

For the OPU site, eddy covariance systems were periodically changed. A sonic anemometer (DA600, Sonic Corp.) was in place from November 2014 to March 2015 and again in November 2015. A different sonic anemometer (Model 81000, R. M. Young, Traverse, Michigan, USA) was in place from March 2015 to April 2015, and a third type of sonic anemometer (Windmaster, Gill Instruments, Lymington, UK) was in place in April 2015. The eddy covariance system was initially a closed-path system using a gas analyzer (LI-6262, LI-COR), until March 2015, and was then changed to an open path system using an open-path infrared gas analyzer (LI-7500, LI-COR). Another eddy covariance system using a sonic anemometer (DA600, Sonic Corp.) and an open-path infrared gas analyzer (LI-7500, LI-COR) was installed on a different edge of the building in November 2015. This additional measurement system increased data acquisition, because we eliminated the data coming from the roof. Consequently, $CO_2$ fluxes were

calculated based on the different systems with relevant corrections. We confirmed that there was no significant difference between open-path and closed-path systems through an inter-comparison (RMSE = 2.18 µmol m$^{-2}$ s$^{-1}$; $F_{open}$ = 1.00 * $F_{closed}$ - 0.03 µmol m$^{-2}$ s$^{-1}$; $R^2$ = 0.84; $F_{open}$ and $F_{closed}$ represent $CO_2$ fluxes by the open and closed paths, respectively), but these flux measurements have higher uncertainties than those from the other sites.

Meteorological and environmental variables were measured at each site. The air temperature, relative humidity, and incoming solar radiation were measured at the three sites. Rainfall, atmospheric pressure, incoming longwave radiation, and ground heat fluxes at the top of the building were measured at OPU. The leaf area index was manually measured approximately once a month using a plant canopy analyzer (LAI-2000, LI-COR) at ten forested sectors in IZM.

The gas analyzers were periodically calibrated. Because the open-path gas analyzer for SAC was installed at a location to which gas cylinders could not be carried easily, we calibrated the analyzer by comparing the signals of $CO_2$ and $H_2O$ densities from a closed-path analyzer (LI-840, LI-COR), whose inlet was located near the open path analyzer. The closed-path analyzer was calibrated every four months using a known $CO_2$ gas, zero $CO_2$ gas and a dew point generator (LI-610, LI-COR). For OPU, the gas analyzers were calibrated at three times in 2015 using the gases and the dew point generator. For IZM, maintenance was regulated, and thus the analyzer was only calibrated at the start and end of the measurements.

*2-3. Data analysis*

In this study, we used one-year eddy covariance data measured in 2015 at SAC and OPU and the period from February 2015 to January 2016 at IZM. Turbulent fluxes were calculated with the eddy covariance method using the Flux Calculator program (Ueyama et al., 2012). Before the half-hourly covariance of vertical wind velocity and scalar quantities were calculated, spike data were removed from the raw data. No trend removal was applied. The artificial fluctuations of sonic air temperature associated with water vapor were corrected. The vertical wind velocity was coordinated as the mean vertical

wind velocity was equal to zero using the double rotation method. The angle-of-attack
errors for the Gill Instruments and R. M. Young anemometers were corrected based on
Nakai and Shimoyama (2012) and Kochendorfer et al. (2012), respectively. The high-
frequency loss for line averaging and sensor separation was corrected using theoretical
transfer functions for the open-path systems (Massman, 2000) and empirical transfer
functions for the closed-path system (Moore, 1986). Air density fluctuations were
corrected based on Webb et al. (1980).
Filtering of the nighttime data using the friction velocity ($u_*$) threshold was not applied
in this study. This was because (1) no clear threshold was obtained in nighttime data, (2)
data coverage at night was small due to the limited flux footprint, and (3) sensible heat
fluxes in the summer months often showed positive values even at night, except for IZM.
Our handling of nighttime data was the same as in previous studies in urban areas (e.g.,
Liu et al., 2012), but a potential underestimate of nighttime fluxes may have occurred.
The storage term was added to the turbulent fluxes for the vegetative site (IZM), whereas
storage was not considered for urban sites (SAC and OPU). The storage term for IZM
was estimated based on $CO_2$ concentrations at the height of the eddy covariance
measurements.
Flux data were selected for each landscape after a quality test and footprint analysis.
First, we applied the quality test to remove half-hourly flux data that included noise based
on a criterion (Appendix B.1 in Ueyama et al., 2012). A stationary test, an integral
turbulence test, and a higher moment test were applied, because flow statistics did not
strongly differ with ideal surfaces (Kaimal and Finnigan, 1994); $\sigma_w/u_*$ at neutral
conditions were 1.3 for SAC, 1.5 for OPU, and 1.3 for IZM; $\sigma_u/u_*$ at neutral conditions
were 2.6 for SAC, 2.6 for OPU, and 3.2 for IZM, where $\sigma_w$ and $\sigma_u$ are the standard
deviation of vertical and horizontal wind velocities, respectively. Half-hourly data were
subdivided into 5 minutes, and then the covariance was calculated for the 5-minute data.
If the difference between the mean of the covariance for the subdivided classes and half-
hourly covariance was greater than 40% of the half-hourly covariance, the data was
rejected as instationary (Foken and Wichura, 1996). We rejected the data when the
turbulent intensity was greater than 50% for IZM and 200% for SAC and OPU of the

intensities predicted by the similarity theory. According to the high moment test (Vickers and Mahrt, 1997), we removed data when the absolute value of skewness was greater than 3.6 or when the value of kurtosis was greater than 14.4. The fluxes, measured when winds came from the tower directions, were also removed. For OPU, the fluxes, measured when winds came from the directions of the building, were also removed. For SAC, based on a footprint model (Kormann and Meixner, 2000), we rejected data, when the source area contributing 80% of the flux footprint contained sea and mountains. Similarly, for IZM, we rejected flux data, when the source area contributing 50% of the flux footprint exceeded the boundary of the urban park. The displacement height was estimated based on MacDonald et al. (1998) for SAC, whereas those for the other sites were estimated at 0.7 times of the mean building or tree heights.

Depending on the wind direction, flux data at IZM and SAC were divided into two data series. For IZA, the flux data from the west represented the urban park, whereas data from other directions represented the rural area consisting of mixed residential and agricultural areas (Fig. 1). For SAC, flux data from the west represented the densely built-up urban center, whereas data from other directions represented the moderate urban to residential area (Fig. 1). Here, we defined the moderate urban area having a green fraction of 27%, which was double that of the dense urban built-up area (Table 1). Consequently, we formed five flux datasets from measurements at the three sites in 2015 for SAC and OPU and in the period from February 2015 to January 2016 in IZM: a dense urban center (west SAC), a moderately urban area (east SAC), a suburb (OPU), an urban park (west IZM), and a rural area (east IZM). Data coverage was 11% in west SAC, 21% in east SAC, 31% in OPU, 16% in west IZM, and 13% in west IZM.

Partitioning $CO_2$ fluxes into gross photosynthetic and respiratory fluxes was conducted only for the west and east IZM and OPU datasets, because the apparent daytime uptake was measured. The flux partitioning was conducted using the Flux Analysis Tool program (Ueyama et al., 2012). First, the relationship between nighttime $CO_2$ fluxes and air temperature was established based on a model (Lloyd and Taylor, 1994). The relationship was determined daily with a 49-day moving window. The gross photosynthetic flux was calculated as the difference between the estimated respiratory flux and the measured $CO_2$

flux. Because the estimated respiratory fluxes consisted of biological fluxes and nighttime
anthropogenic fluxes, it is important to note that the estimated gross photosynthetic fluxes
did not truly represent gross primary productivity, which is often used in ecosystem
studies (e.g., Baldocchi, 2008).
Gaps in the five datasets were filled using the Flux Analysis Tool program. First, small
data gaps for periods of less than 2.0 h were filled by linear interpolation. Second, for the
west and east IZM datasets, gaps were filled using a combination of a look-up-table and
non-linear regression methods (Ueyama et al., 2012), an approach well established for
use in natural ecosystems (Ueyama et al., 2013). For data gaps from the west and east
SAC and OPU, mean diurnal variations were applied, in which a mean diurnal pattern
was created daily using a 51-day moving window. Two mean diurnal patterns were
created, one for weekdays and one for weekends and holidays according to the weekly
cycle.
For evaluating vegetation activity in response to solar radiation ($R_s$), $CO_2$ fluxes ($F_c$)
for IZM and OPU were regressed for summer months using the following rectangular
hyperbola

$$F_c = -\frac{P_{max}\,bR_s}{P_{max}+bR_s} + R_d \qquad\qquad (1)$$

where $P_{max}$ is the maximum photosynthetic rate, b is the initial slope, and $R_d$ is dark
respiration.

*2-4. Upscaling using GIS data*
The annual $CO_2$ flux was upscaled according to the relationship between annual fluxes
and the green fraction. The green fraction was estimated using green census data
developed by the government of Sakai City. The green census data were created using
high-resolution aerial photographs from August 2001, which consisted of polygons of an
approximately 5-m spatial resolution. Based on the high-resolution polygon data, the
green fraction was evaluated at a 500-m spatial resolution. Because the green census data

often classified water as green area, we masked the water area using a land cover data based on a geographical information system (Digital Map 5000 for the Kinki region in 2008 by the Geospatial Information Authority of Japan).

## 3. Results

### 3-1. Meteorological characteristics

The air temperature and vapor pressure deficit (VPD) showed clear seasonal variations (Fig. 2). The air temperature was lowest in January (5.9°C) and highest in August (28.2°C), based on a meteorological station of the Japanese Meteorological Agency. From late July to mid-August, the daily maximum air temperature was continuously higher than 30°C (Fig. 2a). Even in the winter, the daily minimum air temperature often did not reach negative values. The daytime maximum VPD was high from late April to mid-October, but showed a decline in a rainy season, called Baiu, from late June to mid-July, and the typhoon season starting from early September (Fig. 2b). The annual rainfall was 1324 mm $yr^{-1}$ in 2015.

Due to a sea breeze, each site had distinct wind characteristics (Fig. 3). In SAC, winds mainly came from the northwest and east sectors. Winds came from the west and northwest sectors in OPU, and the winds came from the west to north sectors and an east sector in IZM. These characteristics were consistent throughout the seasons (Fig. A1).

### 3-2. Diurnal variations

Diurnal variations at SAC showed greater $CO_2$ emissions during the daytime than at night ($p < 0.01$) (Fig. 4). Daytime emissions were greater in the dense urban center (west SAC) than in the moderately urban area (east SAC) throughout the seasons ($p < 0.01$). Emissions from the urban areas were significantly higher in the daytime than in the nighttime in all seasons ($p \leq 0.01$). Such diurnal variations were similar to those for traffic counts measured by highway exits within the flux footprint (Fig. 4b). Note that the traffic counts at the exits peaked in the evening, whereas those at the entries could peaked in the morning (data not shown). Based on a comparison for diurnal cycles under different weather conditions, $CO_2$ emissions in the afternoon tended to be higher on sunny days

than on rainy or cloudy days for both the west ($p < 0.01$) and east ($p = 0.33$) SAC (Fig.
A2a, b). In contrast to $CO_2$ fluxes, there was no significant difference in the traffic counts
for sunny and rainy/cloudy days.
In contrast to SAC, $CO_2$ fluxes in OPU and IZM showed distinct daytime uptake
especially in summer months (Fig. 4). The magnitude of the daytime uptake was stronger
in the urban park than in the rural area. A daytime uptake was also observed at OPU in
the summer months from April to August. For these three landscapes, the $CO_2$ uptake
increased with solar radiation. According to the rectangular hyperbola regressed between
$CO_2$ fluxes and solar radiation, the rural area ($R^2$=0.46) and urban park ($R^2$=0.34) of IZM
have a stronger light-dependency than the suburb in OPU ($R^2$=0.10). The high light-
dependency in the urban park and the rural area suggests that light was the major
controlling factor in $CO_2$ fluxes at the diurnal timescale. This was consistent with the
smaller $CO_2$ uptake on rainy or cloudy days than on sunny days in the rural area (Fig.
A2e). For the urban park and OPU, the lack of a significant difference among weather
conditions (Fig. A2c, d) suggests that $CO_2$ fluxes were also influenced by other factors,
such as spatial heterogeneity and temperature conditions. For example, sunny days were
warmer in the daytime (approximately 2.5°C in the afternoon) and colder (approximately
1.1°C just before the sunrise) in the nighttime than rainy/cloudy days.

*3-3. Seasonal variations*
Different urban landscapes showed different seasonal variations in the $CO_2$ flux (Fig.
6). Similar to the diurnal variations, distinct biological signals were observed at IZM in
the urban park and rural area. The daily mean $CO_2$ fluxes showed lower emissions with
occasional negative values during summer months in both IZM sites. The suburban site
of OPU generally showed $CO_2$ emissions throughout the seasons, but the emissions rate
tended to be lower in the spring than in other months. The SAC site showed high $CO_2$
emissions throughout the seasons, and higher emissions were observed in the dense urban
center than in the moderately urban area. The seasonal variations in SAC exhibited two
distinct peaks during the summer and winter periods.
The seasonal variations in the daily $CO_2$ flux were dependent on the daily mean air
temperature and exhibited different patterns in different landscapes (Fig. 7). For the urban
site of SAC, $CO_2$ emissions increased as temperatures decreased (0.46-0.27 g C m$^{-2}$ d$^{-1}$
$^{1}$ °C $^{-1}$; $p < 0.1$) when the mean daily temperature was less than 10°C. Higher $CO_2$
emissions were also observed at higher temperatures in SAC. An increase in $CO_2$
emissions at higher temperatures tended to also be observed at OPU ($p = 0.26$). Gas
consumption by university buildings within a footprint of OPU was consistent with the
two seasonal peaks revealing higher consumption in the summer and winter months (Fig.
A3). In the urban park and rural area, $CO_2$ emissions decreased as temperatures increased
above 15°C: -0.27 g C m$^{-2}$ d$^{-1}$ °C $^{-1}$ for the urban park ($p < 0.01$) and -0.13 g C m$^{-2}$ d$^{-1}$ °C
$^{-1}$ for the rural area ($p < 0.01$) when the mean air temperatures were greater than 15°C
(Fig. 7).
Gross photosynthetic fluxes were greater in the summer months than in the winter
months (Fig. 6). Surprisingly, the gross photosynthetic fluxes in the urban park and OPU
were comparable, probably due to the contributions of trees around the university and
from the kofun at OPU. The gross photosynthetic fluxes for the rural area were
approximately half of those for the urban park and OPU. The gross photosynthetic fluxes
for the three sites increased as temperatures increased to more than 20°C at 0.15-0.38 g
C m$^{-2}$ d$^{-1}$ °C $^{-1}$ ($p < 0.01$).

*3-4. Weekly variations*
Among the five landscapes, distinct weekly cycles of $CO_2$ emissions were only
observed at the west SAC and OPU sites (Fig. 8). On average, $CO_2$ emissions on
weekdays were approximately 50% greater than emissions on weekends and holidays ($p$
$< 0.01$) at the west SAC and OPU sites, even though the weekday $CO_2$ flux at the east
SAC was 10% higher than the fluxes on holidays ($p < 0.01$). The greater emissions on
weekdays were consistently observed throughout all seasons, and were consistent with
the traffic counts from the highway exits, where traffic was approximately 23% higher on
weekdays than on weekends and holidays (Fig. 4b).

*3-5. Annual $CO_2$ balance and its spatial variations*

372 All urban landscapes measured in this study acted as net source of $CO_2$ emissions on

373 an annual timescale (Fig. 9; Table 2). The strength of the annual $CO_2$ emissions was

374 negatively correlated with the green fraction ($R^2 = 0.96$; $p < 0.01$); areas with a smaller

375 green fraction had higher annual $CO_2$ emissions. The annual $CO_2$ emissions estimated in

376 this study were lower than those examined using a global synthesis by Nordbo et al.

377 (2012) (Fig. 9).

378 Based on the significant relationship between the green fraction and the annual $CO_2$

379 flux, the annual $CO_2$ fluxes were upscaled to the city scale (Fig. 10). Because the green

380 fraction of Sakai was low in the north and high in the south (Fig. 10a), annual $CO_2$

381 emissions were greater in the north than the south (Fig. 10b). The annual $CO_2$ fluxes from

382 the entire city were 3.3 kg C m$^{-2}$ yr$^{-1}$, which corresponds 0.5 Tg C yr$^{-1}$ or 1.8 Mt $CO_2$ yr$^{-}$

383 $^1$ based on the area of the city (149.81 km$^2$). The estimated emissions were lower than an

384 inventory-based estimate published by the government from 2000 to 2012 (8.0 ± 0.6 Mt

385 $CO_2$ yr$^{-1}$).

387 **4. Discussion**

388 Annual $CO_2$ emissions from Sakai City were in the range of those measured in other

389 studies, but tended to be at the lower end of the range (Fig. 9). For the same fraction of

390 green area (in this case, the green fraction was less than 20%), urban emissions ranged

391 from 4 to 18 kg C m$^{-2}$ yr$^{-1}$ for other cities (Nordbo et al., 2012; Velasco and Roth, 2010).

392 $CO_2$ emission in our city was lower than those measured in urban centers: a dense urban

393 built-up area in London (12.7 kg C m$^{-2}$ yr $^{-1}$; Ward et al., 2015), the historical city center

394 in Florence (8.3 kg C m$^{-2}$ yr$^{-1}$; Gioli et al., 2012), and a residential area of south central

395 Vancouver (6.7 kg C m$^{-2}$ yr $^{-1}$; Christen et al., 2012). The annual emissions in our city

396 were also lower than previous cities that had a similar population density; there were only

397 two cities whose populations were higher than that in our city, but the annual emissions

398 in our city were seventh in the global synthesis (Fig. 12b in Ward et al., 2015). The low

399 $CO_2$ emissions rate in Sakai City was evident in the daytime peaks during the winter

400 months (Fig. 4), compared with a dense urban built-up area in London (e.g., more than

401 50 µmol m$^{-2}$ s$^{-1}$, Ward et al., 2015) and a low built-up area in Beijing (30 µmol m$^{-2}$ s$^{-1}$,

Liu et al., 2012). Warmer winter temperatures (Fig. 2a) may contribute to lower emissions
as a result of reduced building heating and thus lower annual emissions in Sakai City
compared with other northern cities. The annual emissions rate in our urban center was
comparable to those of the densely populated residential areas in Yoyogi, Tokyo (4.3 kg
C $m^{-2}$ $yr^{-1}$, Hirano et al., 2015), and Kugahara, Tokyo (3.4 kg C $m^{-2}$ $yr^{-1}$, Moriwaki and
Kanda, 2004).
The sensitivity of the $CO_2$ emissions to cold temperatures was comparable to that
described in the previous studies (Bergeron and Strachan, 2011; Liu et al., 2012; Pawlak
et al., 2011). The effect of building heating has often been estimated as a slope between
air temperature and the $CO_2$ emissions rate: -2.02 g C $m^{-2}$ $d^{-1}$ $°C^{-1}$ in London (Ward et al.,
2015), -0.21 g C $m^{-2}$ $d^{-1}$ $°C^{-1}$ in Łódź (Pawlak et al., 2011), and -0.35 g C $m^{-2}$ $d^{-1}$ $°C^{-1}$ in
Beijing (Liu et al., 2012). These values are comparable to those obtained in our city: -
0.37 g C $m^{-2}$ $d^{-1}$ $°C^{-1}$ for all SAC ($p = 0.03$) and -0.27 g C $m^{-2}$ $d^{-1}$ $°C^{-1}$ for east SAC ($p <$
$0.01$), when mean air temperatures were less than 15°C (Fig. 7), although the correlation
for west SAC was insignificant. No sensitivities to cold temperatures were found in the
urban park (west IZM), rural area (east IZM), or residential area (OPU), which could be
due to the mixed effects of biological and anthropogenic signals.
$CO_2$ emissions in urban landscapes (SAC and OPU) also increased as temperatures
increased in the summer months (Fig. 7): 0.22 g C $m^{-2}$ $d^{-1}$ $°C^{-1}$ in west SAC ($p = 0.01$),
0.24 g C $m^{-2}$ $d^{-1}$ $°C^{-1}$ in east SAC ($p = 0.02$), and 0.13 g C $m^{-2}$ $d^{-1}$ $°C^{-1}$ in OPU ($p = 0.26$).
The high daytime $CO_2$ emissions were also examined on sunny days when the daytime
air temperature was higher than rainy/cloudy days (Fig. A2). Since traffic did not show a
clear seasonal variation (Fig. 4b), the reason for this increase is unclear, but one
possibility is the contribution of emissions from gas-powered air conditioners (Fig. A3).
The prevalence rate of gas- powered air conditioners is approximately 20% in non-
residential buildings, based on an assessment by the Japan Gas Association. The water
vapor flux in the summer months also significantly increased above a mean daily air
temperature of 17°C (T. Ando, unpublished data), suggesting gas consumption by air
conditioners. Kanda et al. (1997) also measured the high water vapor flux in the summer
at an urban center, Tokyo, and suggested that gas consumption associated with cooling

towers was responsible. In contrast to residences, tall buildings often use gas-based air conditioners, including the Sakai city office and buildings at OPU; especially after the Fukushima nuclear disaster at 2011, nuclear power plants that service the study area do not operate. Consequently, gas-based air conditioners increased (Agency for Natural Resources and Energy, 2015). A weaker dependence in OPU probably occurred because emissions from gas-powered air conditioners from the university building (Fig. A3) were negated by an increase in biological uptake (Fig. 6b). The sensitivity of gross photosynthetic fluxes to warming temperatures was 0.38 g C m$^{-2}$ d$^{-1}$ °C$^{-1}$ in OPU ($p <$ 0.01).

Weekly cycles of $CO_2$ emissions were only observed at urban sites (Fig. 8), representing the strength of human activities. Previous urban $CO_2$ flux studies reported that major contributors to anthropogenic emissions were vehicle emissions and gas consumption (Gioli et al., 2012; Hirano et al., 2015; Velasco et al., 2005; Ward et al., 2013). Velasco and Roth (2010) indicated that weekly cycles were primarily related to vehicle emissions. The traffic count was high on weekdays at SAC (Fig. 4b), and business offices including the university are often more active on weekdays than on weekends and holidays. In contrast, there was no clear weekly cycle in the urban park, and the rural area. Large differences between weekdays and holidays in west SAC and OPU suggest greater contributions of emissions from vehicles and business offices compared with other landscapes. This underscores the importance of temporal variations in $CO_2$ emissions by land use.

The urban park acted as a net annual $CO_2$ source despite the abundant vegetation. Several factors explain the annual emissions from the urban park. First, the urban park frequently suffered from various management activities, such as harvesting and weeding. Such frequent disturbances could decrease the sink and increase source (Gough et al., 2007; Latty et al., 2004). A warmer climate in the urban area may induce higher respiration (Awal et al., 2010). A limited footprint might influence $CO_2$ fluxes arising from emissions from surrounding areas. We re-checked the data selection using stricter criteria according to which we rejected data when 80% of the flux footprint exceeded the boundary of the urban park, but the results were almost the same. Annual $CO_2$ emissions

of 2.4 kg C m$^{-2}$ yr$^{-1}$ were previously measured at an urban park in Germany (Kordowski
and Kuttler, 2010).

Partitioning the flux data measured at a single site with distinct landscapes is a useful

approach in urban flux studies. $CO_2$ fluxes in different landscapes measured at a single
site showed considerably different behaviors (Fig. 4, 6, 9). The approach previously used
for clarifying variations in fluxes in different landscapes involved single flux
measurements (Järvi et al., 2012; Kordowski and Kuttler, 2010; Hirose et al., 2015). The
partitioning concurrently contained the limitations in which data availability decreases
with partitioning. In the study area, sea-breeze circulation was dominant in the summer
months, resulting in a large data gap from certain wind directions (shown in section 2-3).
Accumulating long-term data could be useful for filling the data gap.

The green fraction can be useful for upscaling the annual $CO_2$ flux in urban areas (Fig.

9). The applicability of the green fraction was previously reported based on a global
synthesis based on eddy covariance measurements in urban areas (Nordbo et al., 2012;
Velasco and Roth, 2010; Ward et al., 2015); the green fraction was an index of human
activities (Nordbo et al., 2012). The relationship between the annual $CO_2$ flux and the
green fraction in Sakai City tended to be lower than the relationship revealed by the global
synthesis (Nordbo et al., 2012) (Fig. 9). This difference might indicate that the
relationship differs in each city or country. Other environmental variables, such as
biomass density (Velasco et al., 2016), might improve the scaling of $CO_2$ fluxes in various
cities. Consequently, to quantify the effects of the green fraction on $CO_2$ emissions in
various cities, further direct measurements of $CO_2$ fluxes at various urban sites are
required.

Upscaled annual $CO_2$ fluxes for the city (Fig. 10) were lower than estimated using the

inventory published by the government. According to the inventory, approximately 57%
of $CO_2$ emissions were associated with the industrial sector, but there was no eddy
covariance site in the coastal industrial region. Part of the discrepancy occurred because
our upscaling estimated the net flux of urban emissions and vegetative uptake, whereas
the inventory quantified the emissions. Hirano et al. (1996) estimated that vegetation in
Sakai, primarily in southern sectors, absorbed 0.87 Mt $CO_2$ yr$^{-1}$ of $CO_2$ based on an
inventory-based estimate. Another reason for the discrepancy was that our estimate did
not include hot spot emissions, such as power plants and incineration facilities, or non-
$CO_2$ gas emissions. Oda and Maksyutov (2011) estimated that approximately half of total
annual $CO_2$ emissions were from point sources in most countries. Because our upscaled
$CO_2$ flux did not include such point sources, the $CO_2$ emissions from point sources could
be more rigorously quantified using the governmental inventory than non-point sources
(Oda and Maksyutov, 2011). Thus, the upscaled $CO_2$ flux could be useful as an additional
constraint, providing more information regarding $CO_2$ emissions from non-point sources.
Because our simple method potentially contained uncertainties associated with a limited
number of one-year eddy covariance sites, and only the consideration of the green fraction,
the estimates should be improved with further eddy covariance sites and additional
environmental variables in order to explain $CO_2$ fluxes.

The inherent limitations associated with the eddy covariance method at the urban

environment must be reduced and quantified in future studies. The measurement height
at SAC was more than ten times higher than the mean building height, although reducing
the height was restricted due to sporadic tall buildings. This could induce underestimates
of nighttime fluxes (Oke, 2006), and thus, the annual emission could be underestimated.
$CO_2$ storage within the building was not considered in our study, but must be important
in the late afternoon and early morning (Vogt et al., 2006). In contrast, the measurement
height at OPU was within the roughness sublayer (1.2 to 1.7 times the mean building and
tree heights), and thus fluxes were influenced by localized nearby fields (Oke, 2006).
Separating wind sectors using the footprint analysis may suffer uncertainties when
advection was trigged by wind shifts.

**5.  Conclusion**

Based on continuous measurements using the eddy covariance method at three different

urban sites, the diurnal, weekly, seasonal, and spatial variabilities in the $CO_2$ flux were
evaluated in Sakai, Osaka, Japan. The urban center and university sites acted as $CO_2$
sources in all seasons. A clear weekday/holiday cycle of $CO_2$ emissions was observed at
those sites. A diurnal pattern in the urban center was correlated with those for traffic count.
High emissions were observed in the urban site in both the winter and summer months,
although the traffic did not change seasonally, suggesting that changes in gas consumption
influenced the seasonal variabilities. The urban park and rural area exhibited $CO_2$ uptake
during the summer months, with distinct daytime uptake. Regardless of the green fraction,
all landscapes considered in this study acted as an annual $CO_2$ source. The green fraction
was a useful index that explained the spatial variability in the annual $CO_2$ fluxes, as
suggested in global scale studies (Nordbo et al., 2012; Velasco and Roth, 2010). The
relationship based on eddy covariance data within a single city could be useful to evaluate
$CO_2$ emissions at the city scale. The network of eddy covariance measurements within a
city is useful for characterizing spatial and temporal variations in net $CO_2$ fluxes in urban
areas.

**Acknowledgments**
We thank Dr. Hiroyuki Kaga of Osaka Prefecture University for supporting the GIS
analysis. We thank the people of Sakai City Office for supporting measurements at SAC.
The measurements at IZM were supported by the Sumitomo Foundation (143205). The
measurements at SAC were partly supported by Nissei Foundation grants for
Environmental Problems, H21. Traffic data regarding the Hanshin Expressway were
provided by the Hanshin Expressway Company. Data on gas consumption by Osaka
Prefecture University were provided by the university. We thank two anonymous
reviewers for constructive comments.

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

**Figure and table captions:**
Figure 1. Aerial photograph by Google Earth showing the study area, where the 80% flux
footprint in daytime is shown with red lines. The boundary of Sakai City is shown as a
yellow line.

Figure 2. Seasonal variations of (a) daily mean, maximum, and minimum air temperatures,
(b) daily maximum vapor pressure deficit (VPD) and daily total rainfall. Temperatures
and VPD were measured at 111 m above the ground at the SAC site and rainfall was
measured at the OPU site, during 2015. Temperatures and VPD are shown as a 7-day
running mean.

Figure 3. Relative wind frequency distributions at the three sites during the study period
in 2015. Data are binned in 45° classes.

Figure 4. Mean diurnal variations of (a) $CO_2$ fluxes and (b) traffic count at two highway
exits within the flux footprint of SAC west. The diurnal patterns were created every
consecutive three months in 2015. Because measurements at IZM began in February
2015, diurnal variations for IZM during the period from January to March were
calculated based on data from February and March in 2015 and January in 2016.

Figure 5. Relationships between the $CO_2$ flux and solar radiation measured at (a) the urban
park in IZM, (b) the rural area in IZM, and (c) OPU sites during the period from July
to September of 2015.

Figure 6. Seasonal variations of the daily mean (a) $CO_2$ fluxes and (b) the gross
photosynthetic flux in 2015, shown as 7-day running means.

Figure 7. Relationship between the daily mean air temperature and the daily mean $CO_2$
flux; $CO_2$ flux data were binned at 3°C intervals.

Figure 8. Averaged daily $CO_2$ flux for each day of the week in 2015 for (a) SAC west and
(b) OPU; fluxes for holidays were averaged separately. Vertical lines represent standard
deviation.

Figure 9. Relationship between the annual $CO_2$ flux ($F_{CO2}$) and the green fraction ($f_G$).
The solid line represents a regression based on our flux data for Sakai, and the dashed
line represents a relationship based on a global synthesis (Nordbo et al., 2012).

Figure 10. Spatial distributions of (a) the green fraction and (b) the upscaled net $CO_2$ flux
in Sakai City. The green fraction was calculated at a 500-m spatial resolution based on
an inventory of green spaces.

Figure A1. Relative wind frequency distributions at the three sites during the study period
in 2015 for each season. Data are binned in 45° classes.

Figure A2. Mean diurnal variations of $CO_2$ fluxes at (a) SAC west, (b) SAC east, (c) OPU,
(d) the urban park in IZM, and (e) the rural area in IZM during the period from April
to September. The date are shown as the 1.5-hours running means. Sunny days were
defined as days when the precipitation was less than 5 mm d$^{-1}$, and the daily sum of
solar radiation was greater than 80% of that expected from solar geometry.

Figure A3. Seasonal variations in monthly gas consumption rates at Osaka Prefecture
University for 2015. The data are shown for 16 buildings in the west-sector of the
university, where flux measurements were conducted, and for four buildings located
within the flux footprint.

Table 1. Land cover fraction within d the daytime flux footprint. Landcover classification
was conducted using the Digital Map 5000 for the Kinki region in 2008 by the
Geospatial Information Authority of Japan, and the green space fraction was based on
a green census by the government of Sakai City. Because the land cover classification
and green space are different data sources, the sum of each fraction often exceeds 100%.
The daytime flux footprint was calculated using the analytical footprint model
(Kormann and Meixner, 2000), and median values in 2015 were classified for sixteen
directions (Fig. 1).

Table 2. Annual $CO_2$ fluxes from the eddy covariance measurements and the upscaled
city-scale flux.

Table 1. Land cover fraction within d the daytime flux footprint. Landcover classification was conducted using the Digital Map 5000 for the Kinki region in 2008 by the Geospatial Information Authority of Japan, and the green space fraction was based on a green census by the government of Sakai City. Because the land cover classification and green space are different data sources, the sum of each fraction often exceeds 100%. The daytime flux footprint was calculated using the analytical footprint model (Kormann and Meixner, 2000), and median values in 2015 were classified for sixteen direction (Fig. 1).

| | SAC west | SAC east | OPU | IZM park | IZM rural |
|---|---|---|---|---|---|
| | % | % | % | % | % |
| Residence | 27 | 9 | 9 | 1 | 15 |
| Commercial, industrial, and public offic | 34 | 38 | 69 | 6 | 15 |
| Road | 27 | 29 | 10 | 3 | 6 |
| Green space | 14 | 27 | 44 | 72 | 62 |

Table 2. Annual $CO_2$ fluxes from the eddy covariance measurements and the upscaled city-scale flux.

| Site | $CO_2$ flux $g\ C\ m^{-2}\ yr^{-1}$ |
|------|------|
| SAC west | 4948 |
| SAC east | 3134 |
| OPU | 1270 |
| IZM park | 802 |
| IZM rural | 495 |
| Upscale | 3325 |

# Figure 1

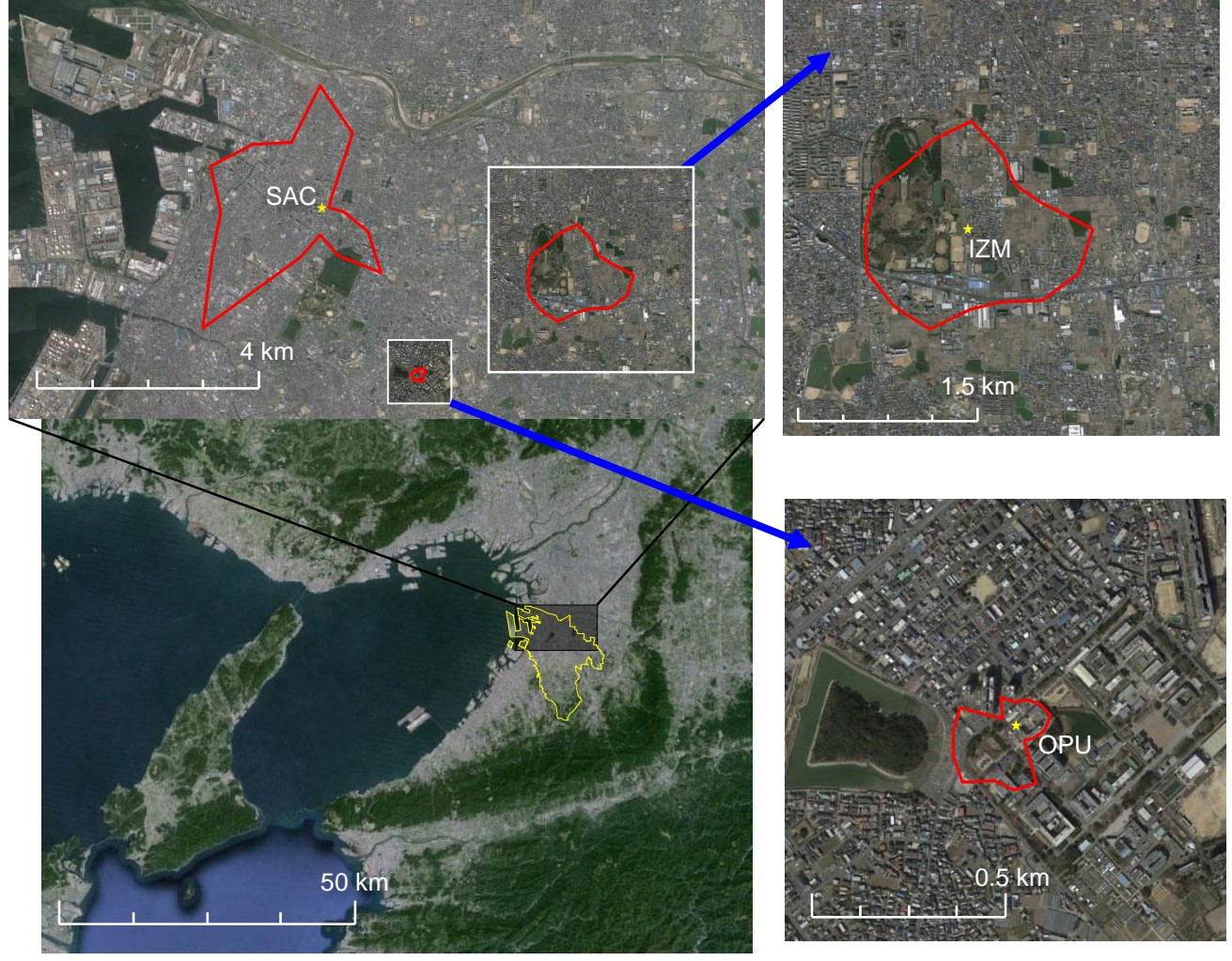

Fig. 1.    Aerial photograph by Google Earth showing the study area, where the 80% flux footprint in daytime is shown with red lines. The boundary of Sakai City is shown as a yellow line.

# Figure 2

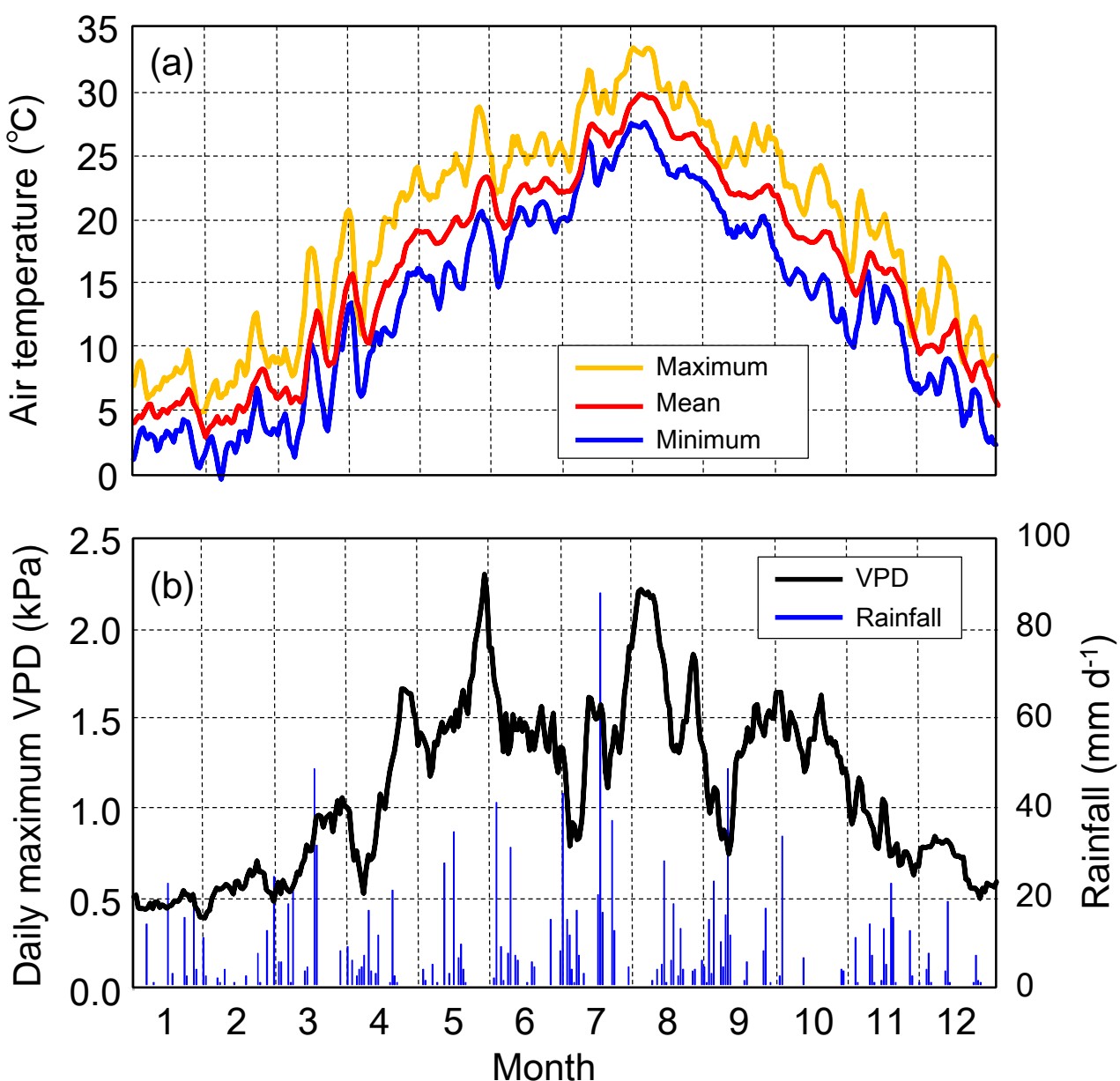

Fig. 2.   Seasonal variations of (a) daily mean, maximum, and minimum air temperatures, (b) daily maximum vapor pressure deficit (VPD) and daily total rainfall. Temperatures and VPD were measured at 111 m above the ground at the SAC site and rainfall was measured at the OPU site, during 2015. Temperatures and VPD are shown as a 7-day running mean.

**Figure 3**

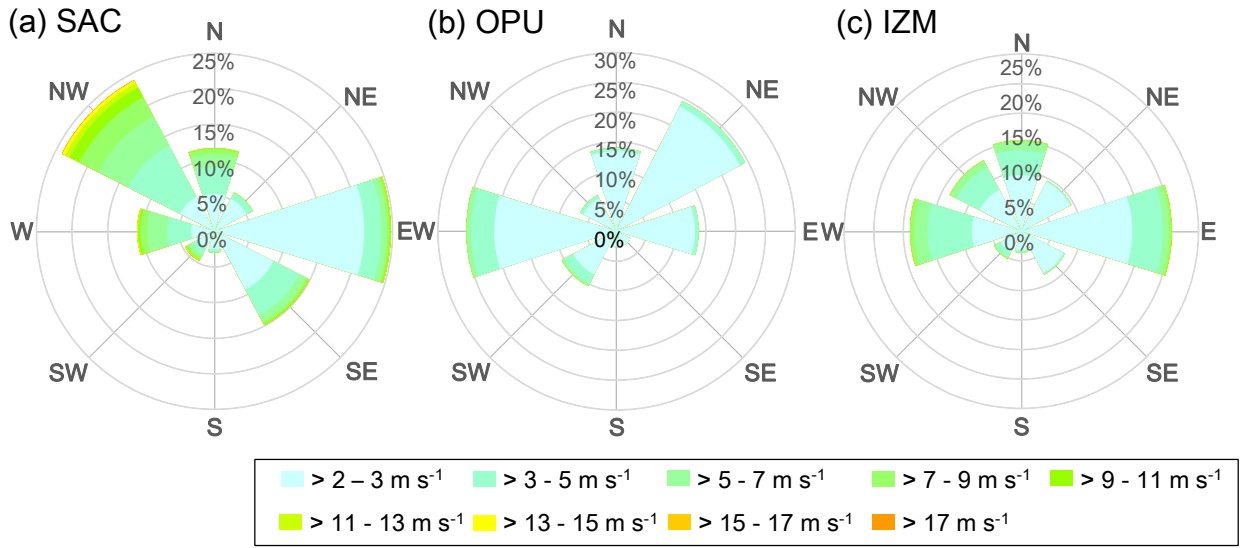

Fig. 3.    Relative wind frequency distributions at the three sites during the study period in 2015. Data are binned in 45° classes.

# Figure 4

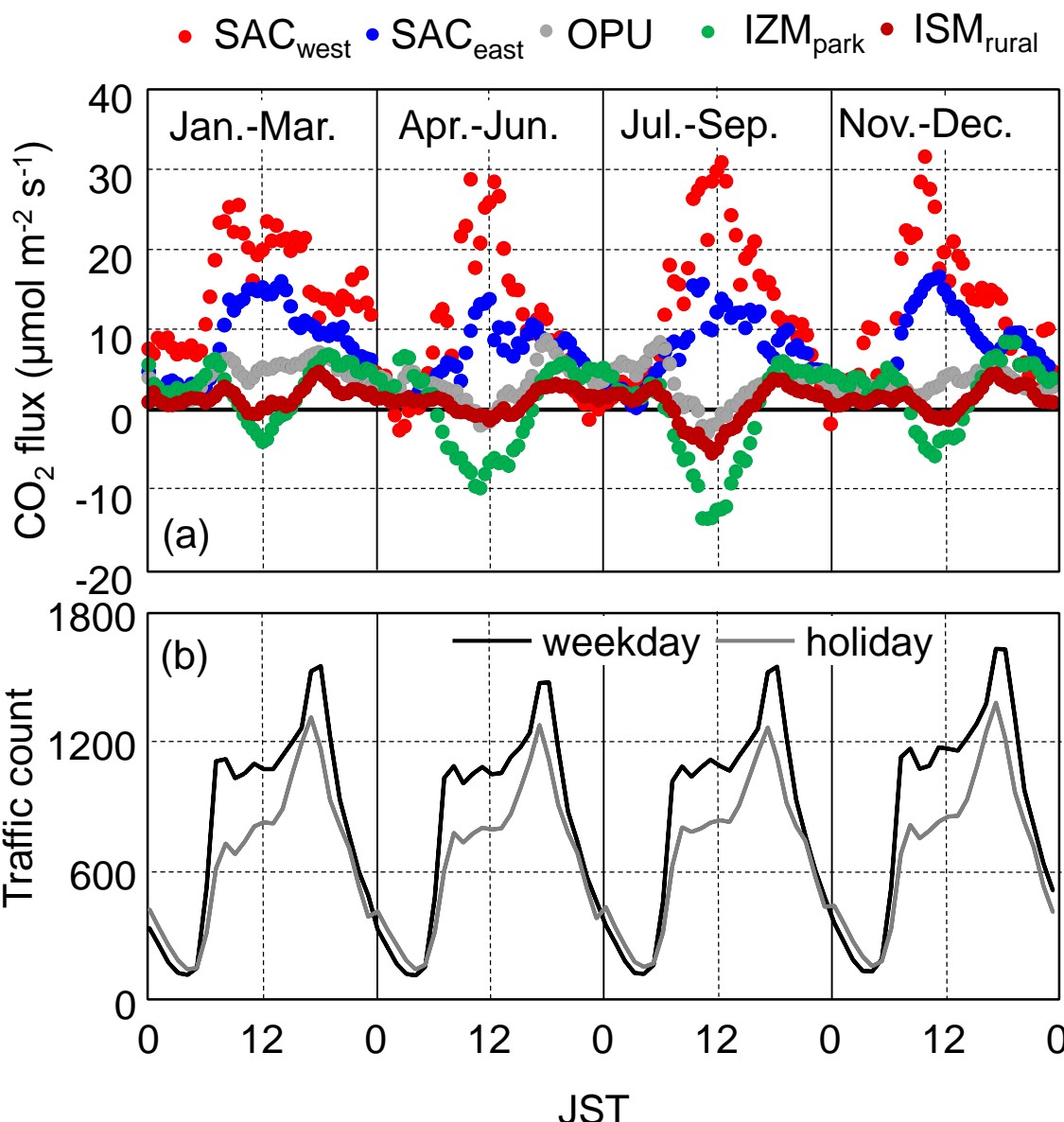

Fig. 4. Mean diurnal variations of (a) $CO_2$ fluxes and (b) traffic count at two highway exits within the flux footprint of SAC west. The diurnal patterns were created every consecutive three months in 2015. Because measurements at IZM began in February 2015, diurnal variations for IZM during the period from January to March were calculated based on data from February and March in 2015 and January in 2016.

# Figure 5

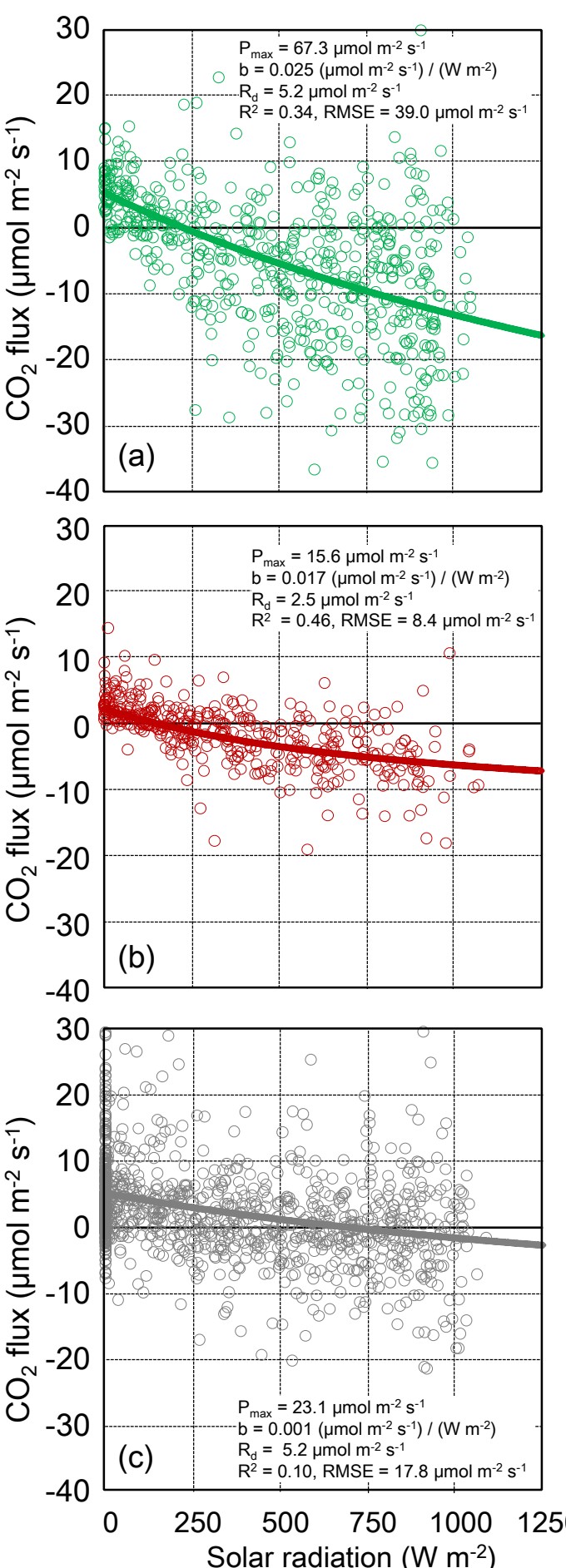

Fig. 5.

Relationships between the $CO_2$ flux and solar radiation measured at (a) the urban park in IZM, (b) the rural area in IZM, and (c) OPU sites during the period from July to September of 2015.

# Figure 6

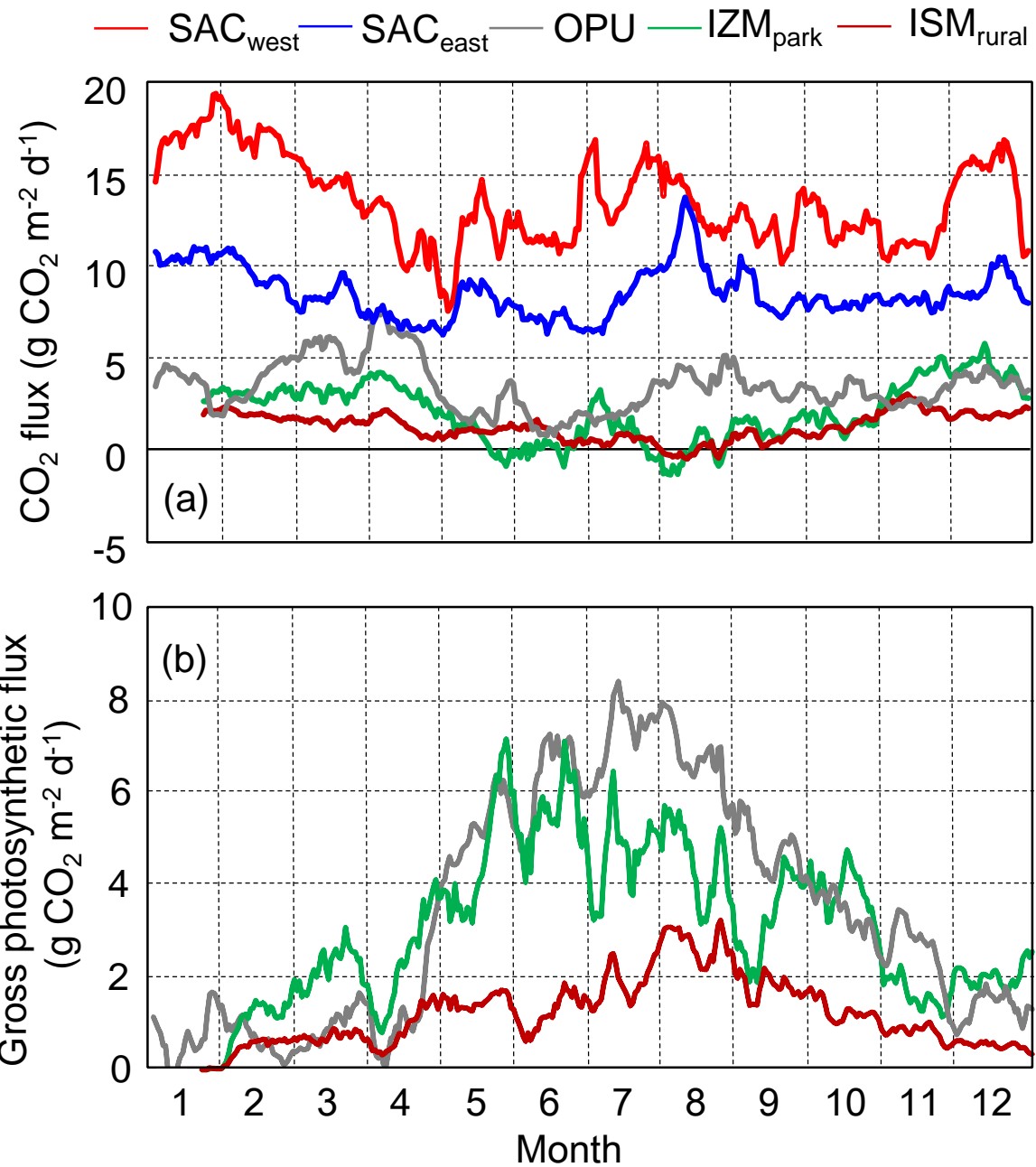

Fig. 6.    Seasonal variations of the daily mean (a) $CO_2$ fluxes and (b) the gross photosynthetic flux in 2015, shown as 7-day running means.

**Figure 7**

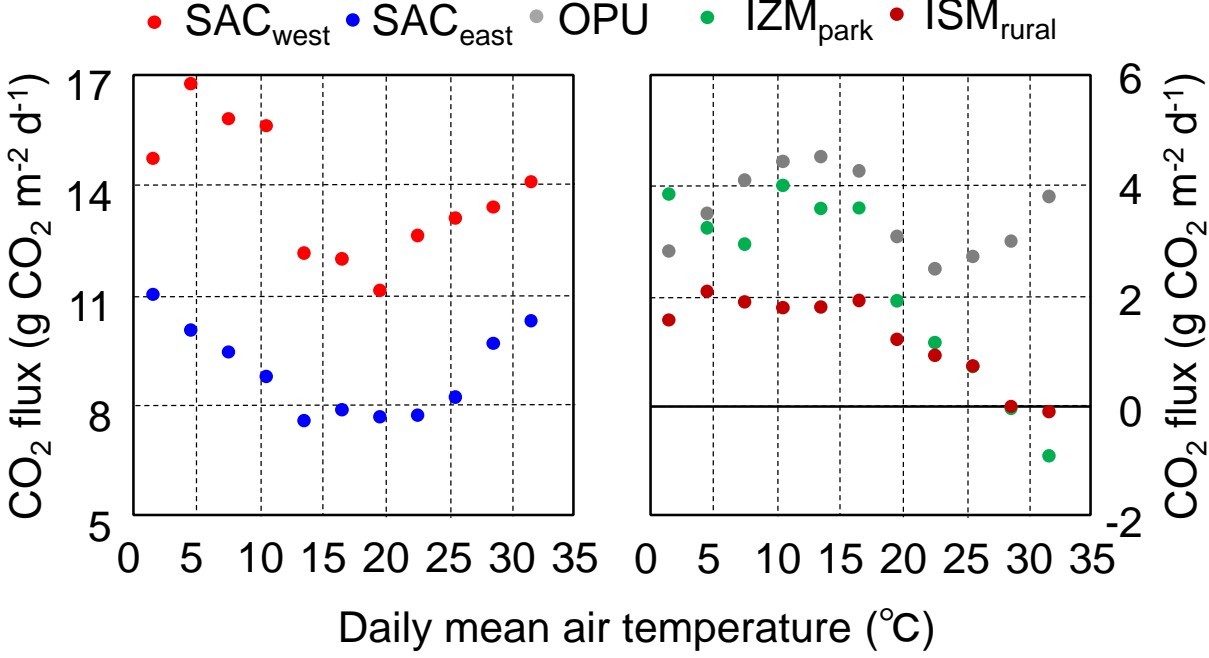

Fig. 7. Relationship between the daily mean air temperature and the daily mean $CO_2$ flux; $CO_2$ flux data were binned at 3° C intervals.

# Figure 8

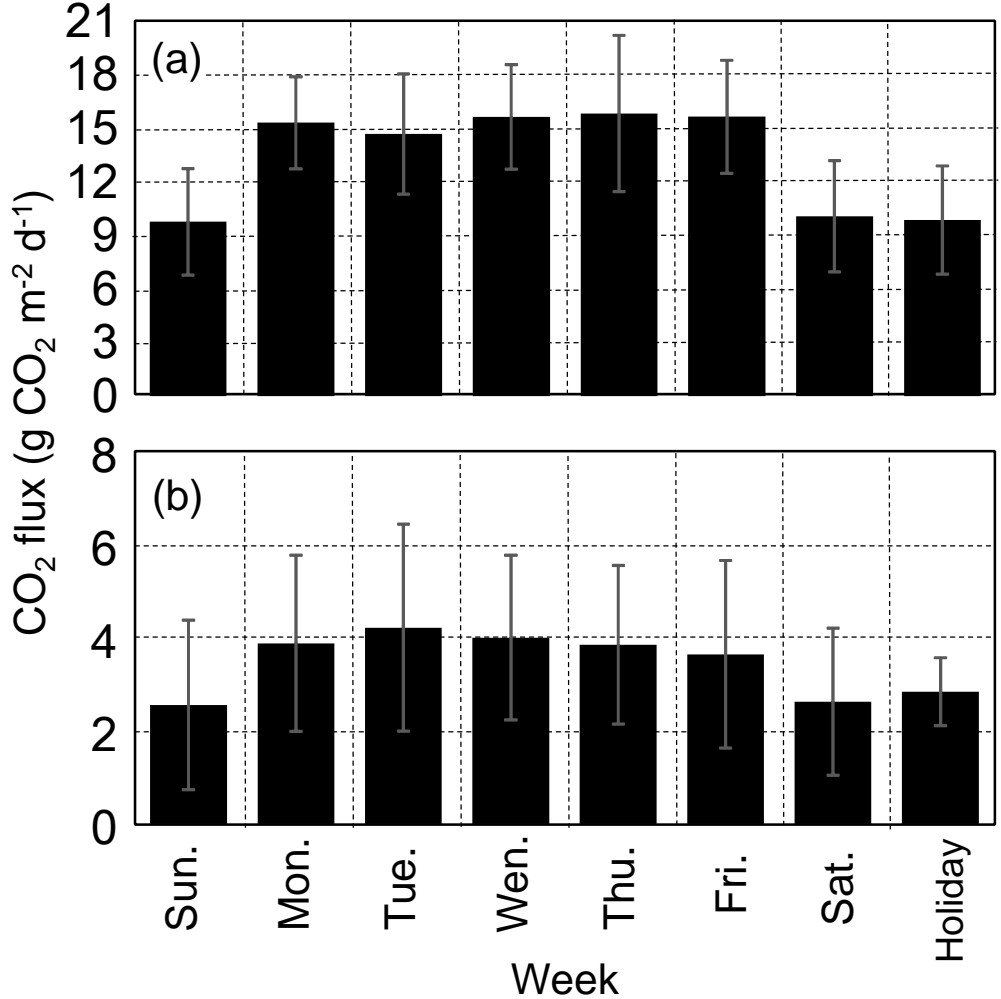

Fig. 8.    Averaged daily $CO_2$ flux for each day of the week in 2015 for (a) SAC west and (b) OPU; fluxes for holidays were averaged separately. Vertical lines represent standard deviation.

**Figure 9**

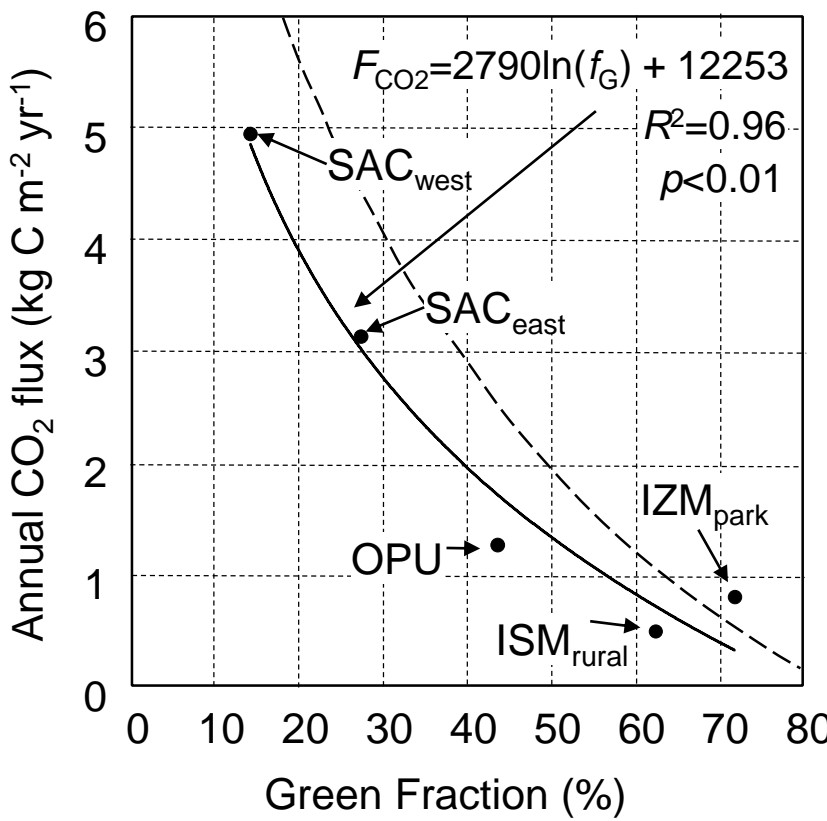

Fig. 9.    Relationship between the annual $CO_2$ flux ($F_{CO2}$) and the green fraction ($f_G$). The solid line represents a regression based on our flux data for Sakai, and the dashed line represents a relationship based on a global synthesis (Nordbo et al., 2012).

**Figure 10**

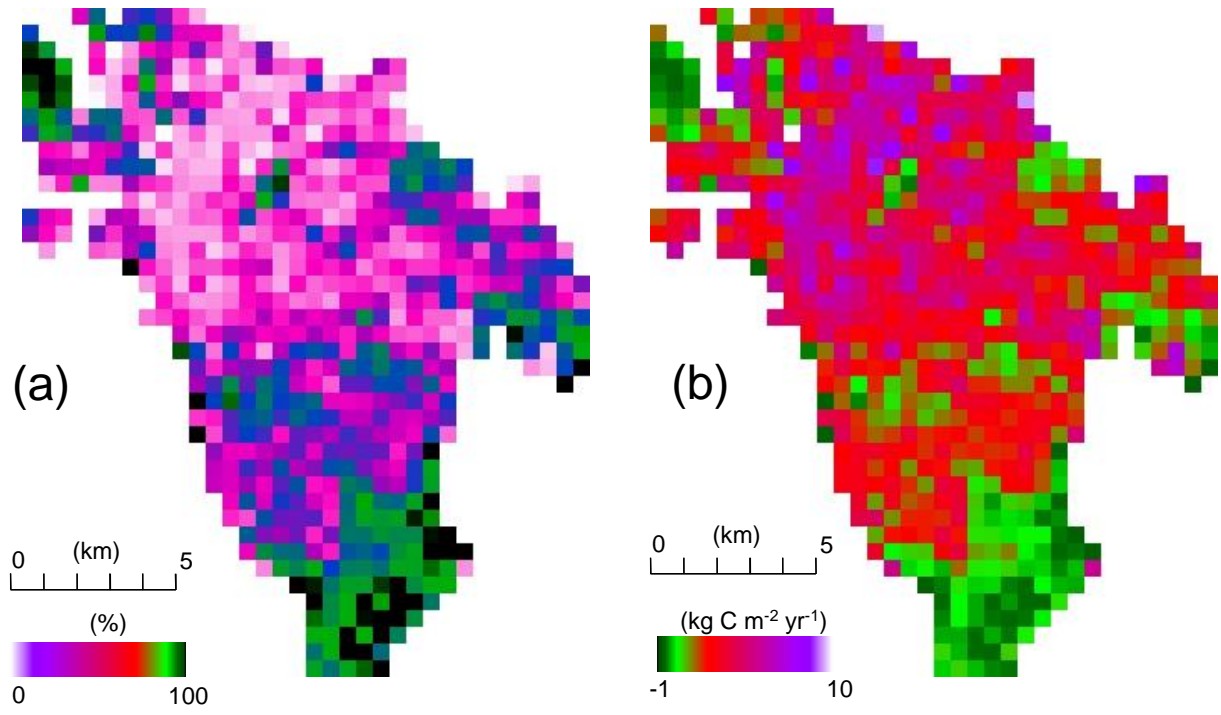

Fig. 10. Spatial distributions of (a) the green fraction and (b) the upscaled net $CO_2$ flux in Sakai City. The green fraction was calculated at a 500-m spatial resolution based on an inventory of green spaces.

# Appendix

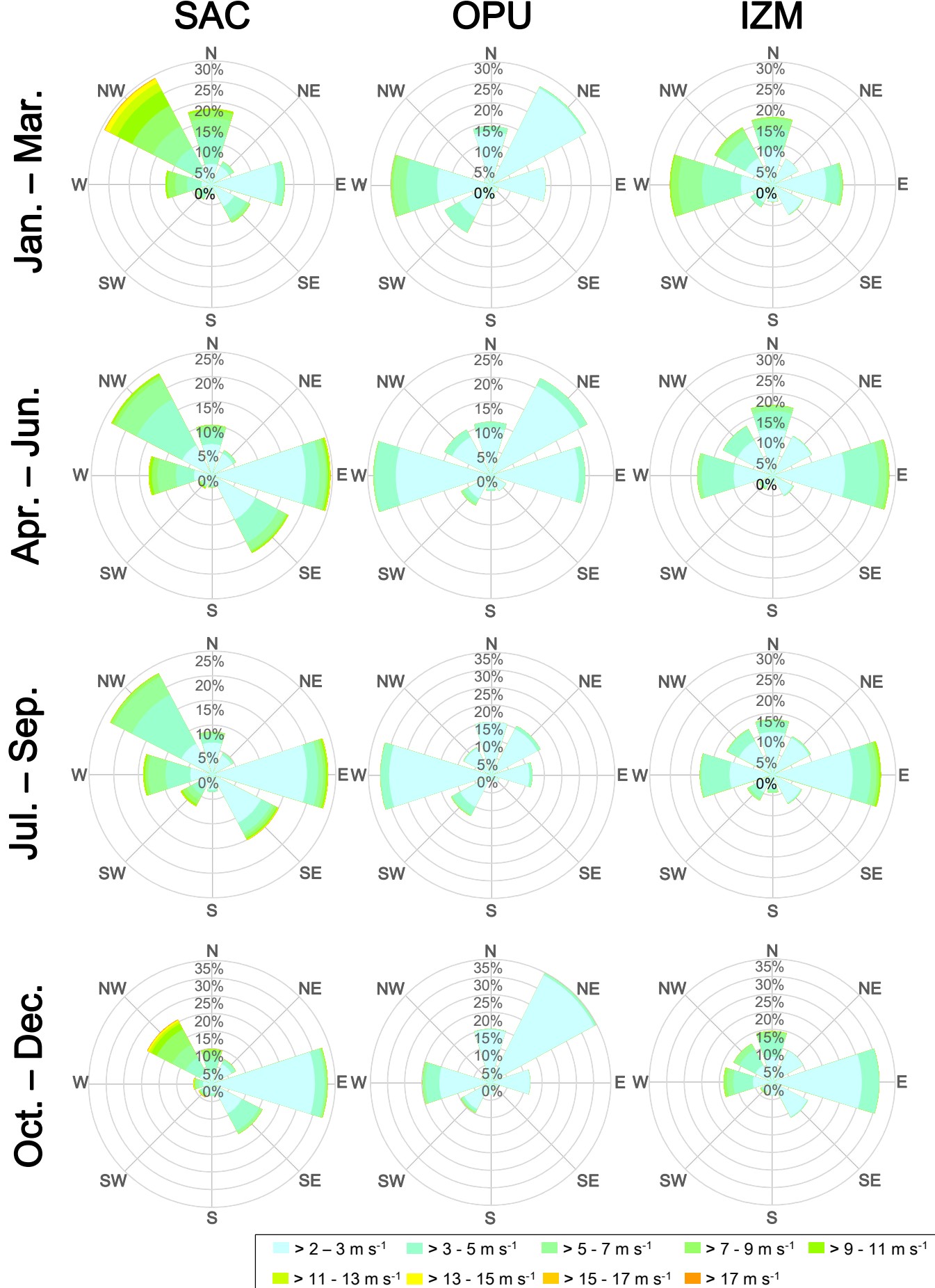

Fig. A1.   Relative wind frequency distributions at the three sites during the study period in 2015 for each season. Data are binned in 45° classes.

# Appendix

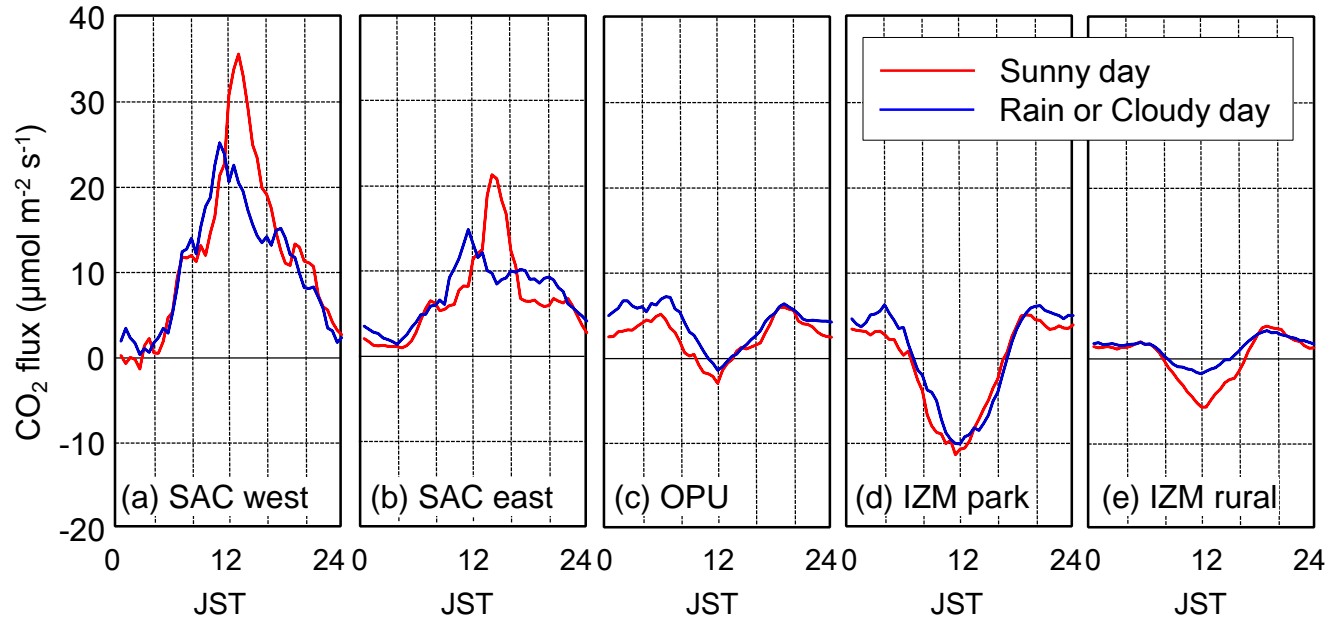

Fig. A2.  Mean diurnal variations of $CO_2$ fluxes at (a) SAC west, (b) SAC east, (c) OPU, (d) the urban park in IZM, and (e) the rural area in IZM during the period from April to September. The date are shown as the 1.5-hours running means. Sunny days were defined as days when the precipitation was less than 5 mm d$^{-1}$, and the daily sum of solar radiation was greater than 80% of that expected from solar geometry.

# Appendix

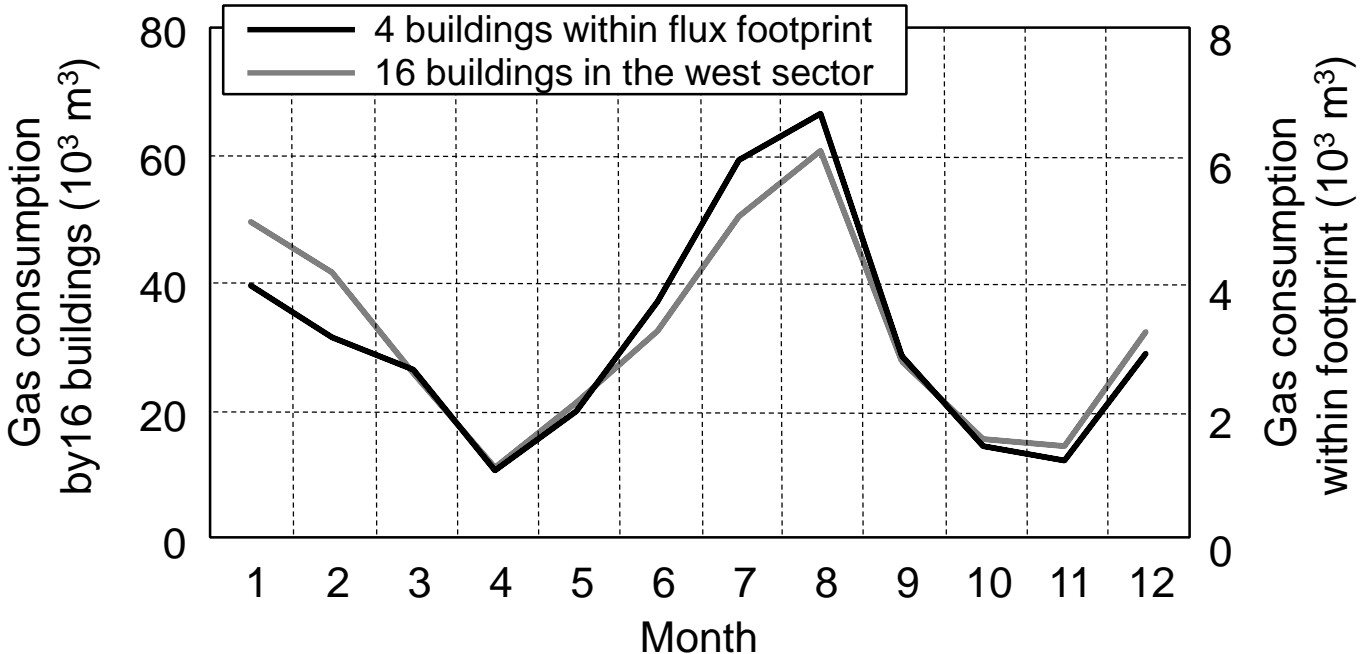

Fig. A3.   Seasonal variations in monthly gas consumption rates at Osaka Prefecture University for 2015. The data are shown for 16 buildings in the west-sector of the university, where flux measurements were conducted, and for four buildings located within the flux footprint.