# Peer review of "Diurnal, weekly, seasonal and spatial variabilities in carbon dioxide flux in different urban landscapes in Sakai, Japan"

_Atmospheric Chemistry and Physics, 2016_

## Referee Comment (RC1) · Anonymous Referee #1 · 13 Jul 2016

An accurate quantification of the greenhouse gas emissions in urban ecosystems relies in the use of multiple techniques according to the particular conditions and needs of the city. Eddy covariance (EC) flux towers have demonstrated to be valuable tools to evaluate fluxes of greenhouse gases. With a proper selection of the footprint they can cover extents of similar size of a complete neighbourhood and help to verify the accuracy of emissions estimated by other techniques.

In this context, Ueayama and Ando instrumented three flux towers in the Japanese city of Sakai during a period of time not indicated in the manuscript with the aim of characterizing the CO2 flux from five different land covers. They used the collected data to extrapolate the observed fluxes to the whole city based on the planar fraction

of the green cover.

Assuming that such approach is appropriate to quantify the total emission from a city, it is not clear how such extrapolation was possible without a comprehensive land-use analysis of the three monitored sites. The manuscript does not provide basic information on the land-use, urban morphology, trees characteristics, vehicular traffic, etc. It is not clear, even, how the observed footprints were estimated without information on the height of the roughness elements (buildings and trees).

Based on the poor description of the methodology, apparently two towers do not meet the basic EC requirement of mounting the instruments within the inertial sublayer. The authors do not present any material to validate the measurements. The description of the data postprocessing is ambiguous. The data quality assurance is not clear. The authors did not consider the criteria of stationarity neither evaluated the turbulence characteristics.

The discussion of the results is pretty qualitative. The results are not solid. Some of their observations are "surprising". For example, they report larger emissions in summer than in winter as a consequence of a major consumption of natural gas for air conditioning. This observation is opposite to all previous studies based on EC in Japan and other mid-latitude cities in the world. This reviewer understand that domestic heating in Japan is by individual heaters. Indeed, they represent an important emission source, but during winter. Air conditioning is also conducted by individual units, but they do not burn fossil fuels. Do buildings in Sakai, perhaps, need to generate their own electricity?

The authors did not follow the basic assumptions of the EC method. A complete understanding of the theory behind is not evident, neither of the advantages and limitations of its application over urban surfaces. The approaches used to explain and extrapolate the observed fluxes are simplistic.

The work presented here does not meet the scientific requirements to be considered

by any peer-reviewed journal.

Specific comments (page/line)

1/13 vegetation activities?

1/14-16 This statement needs an explanation. It is opposite to flux data reported by many previous studies in mid-latitude cities.

1/20-21 This statement is different to the accepted consensus that eddy covariance flux towers are just a tool among others to evaluate urban emissions. The method has advantages and limitations. Multiple methods are needed to quantify emissions for a comprehensive climate change mitigation assessment.

1/27 Inventories of what?

2/9 What is the difference between an urban park and an urban forest?

2/10 Results ... None result is discussed in this paragraph.

2/12-13 The role of vegetation in the CO2 exchange depends on the landscape characteristics and anthropogenic activities. In urban districts, all studies based on eddy covariance flux measurements have found that vehicular traffic and domestic heating are the main contributors. Depending on the extension of the green areas and trees characteristics, urban vegetation may be relevant.

2/13 It is not correct to affirm that vegetation fraction and anthropogenic activities are correlated. Those studies analyzed the correlation between the planar fraction of green space and CO2 flux in residential neighborhoods. The biogenic component (vegetation + soil), human respiration, vehicular traffic and domestic consumption of fossil fuels have been reported as the main contributors for that particular type of urban land-use.

2/23 Define moderate urban area.

3/4 Table 1 does not provide information on the characteristics of the monitored sites.

3/16 What does the category of others include? It is the second largest component.

3/21 Ancient tumulus?

3/26-27 It means that that the instruments were mounted at 127 m above the ground, 12 times the average height of the roughness elements (i.e. buildings). Were the instruments mounted within the inertial sublayer? Fluxes obtained from very tall towers (> 100 m) may not be representative of the urban landscape. Tall towers may reach above the top of the collapsing boundary layer at night, causing difficulties in interpreting the flux data.

3/28 Avoid grey literature.

3/31-32 The inertial sublayer is usually located at a height of 2-4 times the mean height of the buildings. Instruments mounted at 2 m over a building of 16.2 m are clearly not placed within such layer.

4/11-12 The manuscript does not provide any material to support this observation.

4/21 No trend removal was applied, why?

4/29 If no filtering based on u* was applied, was the stationarity criteria used? (see Aubinet et al. Advances in Ecological Research 30, 113-175, 2000).

4/29 Was the flux storage below the towers during night time considered?

5/1-8 The data quality assurance is poorly described.

5/15 The percentages of data coverage are useless if the total measurement period for each tower is not provided.

5/18 $CO_2$ sequestration or carbon capture are common terms to talk about negative fluxes. Assimilatory flux is not a common term in the community of urban climatologists.

5/18 Are there no trees at the SAC sites?

5/19 What does the biological signal consider? Does it include the soil respiration

contribution?

6/5 This reviewer does not consider such a simplistic approach can solve, even roughly, the potential role of vegetation in the urban CO2 flux. Weissert et al. Urban Climate 8, 100-125, 2014 provide a comprehensive review of methods to quantify the potential carbon sequestration by urban vegetation. Check also the approaches followed by Velasco et al. Atmos. Chem. Phys. 13, 10185-10202, 2013 and Ward et al. Atmos. Chem. Phys. 13, 4645-4666, 2013 to evaluate the urban vegetation contribution.

6/14 Biomass density is the important parameter to investigate (see Velasco et al. Landscape and Urban Planning 148, 99-107, 2016).

7/15 CO2 contributions from tombs? Do you suggest that corpses capture carbon?

7/18 Are statistically different the weekly variations? Any difference between fluxes on Saturdays and Sundays?

7/25 Table 2 was not included in the manuscript.

7/29-31 This sentence contradicts the previous statement that vegetation is correlated with anthropogenic activities (2/15).

7/31 A comparison with flux data reported by Hirano et al. SOLA 11, 100-113, 2015 and Moriwaki and Kanda Journal of Applied Meteorology 43, 1700-1710, 2004 for two residential neighborhoods of Tokyo is needed.

8/7-9 A proper comparison should consider differences on the urban morphology, climatology, population density, anthropogenic activities, etc.

8/11 Be consistent with the use of units.

8/20 Such slopes cannot be appreciated in Fig. 4.

8/20-21 The green dots in Fig 4(b) show a clear variation with temperature, at least in the 10-35 degC range.

9/10 Migration of urban emissions to urban parks?

9/20-21 Indeed, a comprehensive partitioning of the CO2 flux as a function of the upcoming wind direction may be useful to understand differences according to the surface characteristics. However, such partitioning may suffer of severe uncertainties due to advection issues triggered by wind shifts during the averaging periods.

9/29 Green fraction as an index of human activities?

10/2-3 This is a reason why eddy covariance flux towers cannot solve by themselves the puzzle of the greenhouse gas emissions in urban ecosystems.

10/3-4 Such comparison should consider the extrapolated flux after subtracting the "biological" component.

---

## Author Comment (AC1) · 19 Jul 2016

We thank the referee for the helpful comments. Please find attached our reply.

Please also note the supplement to this comment:
http://www.atmos-chem-phys-discuss.net/acp-2016-334/acp-2016-334-AC1-supplement.zip

---

## Referee Comment (RC2) · Anonymous Referee #2 · 15 Oct 2016

Title: Diurnal, weekly, seasonal and spatial variabilities in carbon dioxide flux in different urban landscapes in Sakai, Japan By Masahito Ueyama, Tomoya Ando

General comment: This study presents flux of CO2 measured at three different types of locations (urban suburb and rural) in Sakai, Osaka. The fluxes have been calculated using the eddy covariance method. In summary, the paper presents important result about the variations of flux in different time scales. The data looks reliable and worth reporting. However, I am not fully satisfied with the discussion which is largely qualitative and very brief. Therefore, there is a significant scope to improve the draft considering following aspects.

(1) The experimental uncertainties in the measurements of eddy parameters (CO2,

winds, etc.) have not been reported. In the "Observations" section, the errors and calibration procedure should be presented in details. The variations of meteorological parameters (wind parameters, RH, temp) should be presented (Figures, wind rose) for each season.

(2) The authors should work to make proper statistical representations of results (using mean, median, percentiles, standard deviation, etc.). None of the representations (figures) used in the paper show the variance of $CO_2$ flux on daily, weekly and monthly scales. Accordingly, the discussion is "overall" but not the "detailed".

(3) About the results and discussion, sometimes I am confused to see overlaps between the interpretations of data for different sites. Hence, the important governing processes at each site. This is because the discussion is not structured and very brief. For example, the results of "3.1 Diurnal variations" "3.3 Weekly variations" at all sites have been summarized just in few lines. Therefore, it is difficult to follow and appreciate the discussion.

(4) Additional analysis: It would be interesting to see how the diurnal flux changes with weather conditions in each season. For this, I suggest to separate the data, at least for rainy, cloudy and clear-sky days in each season. Please provide the diurnal flux figures measured under distinct weather conditions (rainy, cloudy and clear sky)

(5) English should be improved, sometimes choice of word and phrase are not appropriate.

(6) Overall, the paper looks a kind of well written report. However, a scientific paper requires more detailed representations of both results and discussion.

Some specific comments are given here:

Page 1 Line 14-16: Following sentence is ambiguous and needs to be re-written. "In contrast, the dense and moderately urban areas exhibited higher emissions in winter and summer months, when emissions significantly increased as air temperature

increased in summer and air temperature decreased in winter."

Page 1 Line 25: In this sentence, I do not find the logic to use "Consequently, ..."

Page 1 Line 27: " Global CO2 emissions have often been estimated using inventories ..." I do not understand this, what do you mean by "inventories" here ?

Page 2 Line 15: "because vegetation fraction can be correlated with anthropogenic activities. " How?, an explanation is required.

Page 3 line 1: " according to the apanese Meteorological " Spelling issue, please correct

Page 4 line 19: Following sentence needs correction "using" has come twice

Turbulent fluxes were calculated using the eddy covariance method using the Flux Calculator program

Page 6 " 3.1 Diurnal variations" The discussion in this section is very qualitative. It is needed to be more quantitative in terms of site to site variations represented by suitable diurnal statistical analysis?

Page 6 3.2 Seasonal variations Again, the discussion in this section is very qualitative. It is needed to be more quantitative in terms of site to site variations represented by suitable seasonal statistical analysis?

Page 6 Line 22-23, Previous studies ...............our city. This sentence is not clear, what authors wish to convey?
* * *

---

## Author Response (AR1)

**Responses to reviewer's comments 1.**

Ueyama and Ando
Diurnal, weekly, seasonal and spatial variabilities in carbon dioxide flux in different urban landscapes in Sakai, Japan

**1.** *An accurate quantification of the greenhouse gas emissions in urban ecosystems relies in the use of multiple techniques according to the particular conditions and needs of the city. Eddy covariance (EC) flux towers have demonstrated to be valuable tools to evaluate fluxes of greenhouse gases. With a proper selection of the footprint they can cover extents of similar size of a complete neighbourhood and help to verify the accuracy of emissions estimated by other techniques.*
*In this context, Ueayama and Ando instrumented three flux towers in the Japanese city of Sakai during a period of time not indicated in the manuscript with the aim of characterizing the CO2 flux from five different land covers. They used the collected data to extrapolate the observed fluxes to the whole city based on the planar fraction of the green cover.*

> Thank you very much for your understanding of our research background, careful editing and very useful comments. We have revised our manuscript based on the two reviewers' comments. In addition, the revised manuscript has been edited using a NPG Language editing service. All revisions for reviewer 1 are marked as red and revisions for reviewer 2 are marked as green, in the revised manuscript. The edited by the NPG Language editing service was marked as purple. The study period is one year at 2015, and we have add this information in the revised manuscript.

**2.** *Assuming that such approach is appropriate to quantify the total emission from a city, it is not clear how such extrapolation was possible without a comprehensive land-use analysis of the three monitored sites. The manuscript does not provide basic information on the land-use, urban morphology, trees characteristics, vehicular traffic, etc. It is not clear, even, how the observed footprints were estimated without information on the height of the roughness elements buildings and trees).*

> The upscaling method that we used is based on the published method synthesizing the eddy covariance data from various cities (Nordbo et al., 2012), where the green fraction is a single scaling index explaining spatial variabilities in urban fluxes. In this study, we examined the applicability of the method for our city, and found that our data and analyses were consistent to the previous findings. We agree that upscaling the eddy covariance fluxes into city-scale

had considerable uncertainties. Thus, we have added sentences for limitation and possible improvement of our approach in discussion section in Lines 480-482, and Lines 500-503. Although considerable uncertainties were contained, we believe that the upscaling using the green fraction was the important first step as suggested by recent publications (Nordbo et al., 2012; Velasco and Roth, 2010; Ward et al., 2015).

We agree that the basic information of the study site is important, and have added this information, such as building and tree heights in Lines 109-112 for SAC; in Lines 124-126 for OPU; and in Lines 135-137 for IZM; land cover for IZM in Lines 121-122; and population density for rural IZM in Lines 128-129. We have also added the sentence how building and tree heights were used for estimating footprint as "The displacement height was estimated based on MacDonald et al. (1998) for SAC, whereas those for the other sites were estimated at 0.7 times of the mean building or tree heights." in Lines 231-233. Traffic count and gas consumption data were shown in the manuscript, to explain the temporal variabilities in $CO_2$ fluxes, although the cities-scale traffic and gas consumption data were unfortunately publically available. Tree characteristics for the urban park were very complicated with various species, such as *Quercus*, *Prunus*, *Cerasus*; there were no dominant species in the park. Consequently, we cannot list those species in the manuscript.

**3.** *Based on the poor description of the methodology, apparently two towers do not meet the basic EC requirement of mounting the instruments within the inertial sublayer. The authors do not present any material to validate the measurements. The description of the data postprocessing is ambiguous. The data quality assurance is not clear. The authors did not consider the criteria of stationarity neither evaluated the turbulence characteristics.*

Yes, the descriptions for our study sites were insufficient, and we have added further site information including site characteristics and data processing; study period in Lines 186-187, storage term estimation in Lines 206-209, flow statistics in Lines 212-217, quality control in Lines 217-225, displacement height estimation in Lines 213-233, calibrations in Lines 175-183, and meteorological and wind conditions in Lines 287-300 and new Fig. 2, 3, and A1. Details in each revision are shown in answers for the specific comments below.

We believe that the measurements in SAC were conducted within the inertial sublayer by following reasons. The mean building height of SAC was 10.7 ± 3.1 m, but the building heights were highly skewed in low buildings, resulting in that it was difficult for interpreting the arithmetic mean. The mean building height of buildings greater than 20 m, which occupy 33% of total building area, were 36 m. Those tall buildings prevent the flux measurements at lower heights, because $CO_2$ sources, such as gas engines for air conditioner, were often

located at the roof of tall buildings. For OPU, the measurements were conducted at the roughness sublayer; the mean height of buildings and trees for the upwind directions were 10.3 m and 13.1 m respectively. We have added those limitations in the revised manuscript in the final paragraph in the discussion section in Lines 504-514. We believe that our flux measurements contributes urban flux study, even though uncertainties are contained.

We checked the general flow statistics: e.g., σw/u* at neutral conditions were 1.3 for SAC, 1.5 for OPU, and 1.3 for IZM; σu/u* at neutral conditions were 2.6 for SAC, 2.6 for OPU, and 3.2 for IZM, which did not strongly differ with those examined at bare soils (Kaimal and Finnigan, 1994), urban (Högström et al., 1982), or a forest (Ueyama et al., 2014). We have added this information in the method section in Lines 212-217 as "A stationary test, an integral turbulence test, and a higher moment test were applied, because flow statistics did not strongly differ with ideal surfaces (Kaimal and Finnigan, 1994); $\sigma_w/u_*$ at neutral conditions were 1.3 for SAC, 1.5 for OPU, and 1.3 for IZM; $\sigma_u/u_*$ at neutral conditions were 2.6 for SAC, 2.6 for OPU, and 3.2 for IZM, where $\sigma_w$ and $\sigma_u$ are the standard deviation of vertical and horizontal wind velocities, respectively.".

**4.** *The discussion of the results is pretty qualitative. The results are not solid. Some of their observations are "surprising". For example, they report larger emissions in summer than in winter as a consequence of a major consumption of natural gas for air conditioning. This observation is opposite to all previous studies based on EC in Japan and other mid-latitude cities in the world. This reviewer understand that domestic heating in Japan is by individual heaters. Indeed, they represent an important emission source, but during winter. Air conditioning is also conducted by individual units, but they do not burn fossil fuels. Do buildings in Sakai, perhaps, need to generate their own electricity?*

Thank you for pointing out the issues. Yes, our discussion for $CO_2$ emissions for the summer was insufficient. High $CO_2$ emissions in the summer could be a unique nature of dense urban built-up in Japanese big cities in recent years. As you know, unclear power plants are no longer available in most of Japanese cities after the Fukushima nuclear disaster at 2011 by public opinion. Consequently, gas-based air conditioners are increasing currently for industrial and commercial buildings expect residences. We have added this discussion in the revised manuscript in Lines 427-436 as "The prevalence rate of gas- powered air conditioners is approximately 20% in non-residential buildings, based on an assessment by the Japan Gas Association. The water vapor flux in the summer months also significantly increased above a mean daily air temperature of 17°C (T. Ando, unpublished data), suggesting gas consumption by air conditioners. Kanda et al. (1997) also measured the high

water vapor flux in the summer at an urban center, Tokyo, and suggested that gas consumption associated with cooling towers was responsible. In contrast to residences, tall buildings often use gas-based air conditioners, including the Sakai city office and buildings at OPU; especially after the Fukushima nuclear disaster at 2011, nuclear power plants that service the study area do not operate. Consequently, gas-based air conditioners increased (Agency for Natural Resources and Energy, 2015).". For supporting the discussion, we showed the gas consumption data (Figure S1), where the summer gas consumption was higher than those in the winter. In the university, before the Fukushima nuclear disaster, gas-based air conditioner was not used.

**5.** *The authors did not follow the basic assumptions of the EC method. A complete understanding of the theory behind is not evident, neither of the advantages and limitations of its application over urban surfaces. The approaches used to explain and extrapolate the observed fluxes are simplistic.*

We agree that our eddy covariance measurements contained uncertainties, and have added a new paragraph showing our limitation in Lines 504-514 as "The inherent limitations associated with the eddy covariance method at the urban environment must be reduced and quantified in future studies. The measurement height at SAC was more than ten times higher than the mean building height, although reducing the height was restricted due to sporadic tall buildings. This could induce underestimates of nighttime fluxes (Oke, 2006), and thus, the annual emission could be underestimated. $CO_2$ storage within the building was not considered in our study, but must be important in the late afternoon and early morning (Vogt et al., 2006). In contrast, the measurement height at OPU was within the roughness sublayer (1.2 to 1.7 times the mean building and tree heights), and thus fluxes were influenced by localized nearby fields (Oke, 2006). Separating wind sectors using the footprint analysis may suffer uncertainties when advection was trigged by wind shifts.".

Our analysis and scaling method might be simple, but this simple scaling method was previously examined as a good method at the global scale (Nordbo et al., 2012). We believe that the locations of SAC and IZM were the best location for estimating the eddy fluxes within our city, although the measurements were not conducted at a perfectly ideal location. Owing to current available number of eddy flux data at urban areas, we believe that our data and results could contribute urban flux studies for charactering dense urban built-up, urban park, and rural landscapes for a mid-latitude city, and meet the current standard of the urban flux studies. We believe that our results could progress current urban fluxes studies by showing temporal and spatial variabilities in $CO_2$ fluxes at dense built-up urban center, rural area, and urban park within a single city with available data of traffic, gas consumption, and

GIS.

**6.** *The work presented here does not meet the scientific requirements to be considered by any peer-reviewed journal.*

We agree that previous manuscript lack important information, statistics for results and discussion, and description of uncertainties, but believe that the revised manuscript meets the scientific requirements. We truly thank reviewer's comments for improving our manuscript.

*Specific comments (page/line)*
*1/13 vegetation activities?*

We have changed the term to "photosynthetic uptake".

*1/14-16 This statement needs an explanation. It is opposite to flux data reported by many previous studies in mid-latitude cities.*

As described in answer for the general comment 4, we have added further discussion in the revised manuscript as the answer in the general comment.

*1/20-21 This statement is different to the accepted consensus that eddy covariance flux towers are just a tool among others to evaluate urban emissions. The method has advantages and limitations. Multiple methods are needed to quantify emissions for a comprehensive climate change mitigation assessment.*

We agree that the statement was not valid, and have revised the sentence in Lines 39-42 as "A network of eddy covariance measurements is useful for characterizing the spatial and temporal variations in net $CO_2$ fluxes from urban areas. Multiple methods would be required to evaluate the rationale behind the fluxes and overcome the limitations in the future.".

1/27 Inventories of what?

The global inventories that we cited are based on combination of point source databases, statistics for national and regional $CO_2$ emissions, and satellite remote sensing (Oda and Maksyutov); we have added the information in the revised manuscript in Lines 53-55.

*2/9 What is the difference between an urban park and an urban forest?*

The urban park is a mixture of managed trees, grassland, and human architectures and used for human recreation, whereas urban forest in the literature is a forest in a city and was not used for human recreation.

*2/10 Results : : : None result is discussed in this paragraph.*

We have added sentences for knowledge from those previous researches in Lines 70-76 as "These results have indicated that cities emits a considerable amount of $CO_2$ into the atmosphere from human activities, such as vehicle traffic and household heating in the wintertime. Even in urban parks, $CO_2$ was emitted to the atmosphere due to human activities (Kordowski and Kuttler, 2010). The magnitude of $CO_2$ emissions and its temporal variability depended on the city, associated with the type of human activities under different climate conditions (Järvi et al., 2012; Moriwaki et al., 2006; Velasco et al., 2016; Ward et al., 2013, 2015), and the role of urban vegetation (Awal et al., 2010; Kordowski and Kuttler, 2010; Peters and McFadeen, 2012; Ward et al., 2015), showing considerable heterogeneities.".

*2/12-13 The role of vegetation in the CO2 exchange depends on the landscape characteristics and anthropogenic activities. In urban districts, all studies based on eddy covariance flux measurements have found that vehicular traffic and domestic heating are the main contributors. Depending on the extension of the green areas and trees characteristics, urban vegetation may be relevant.*

We agree that vegetation was not control the spatial variabilities in $CO_2$ flux. We have revised the sentence in Lines 77-81 as "Multi-site eddy covariance towers were used to synthesize the data and showed that green fraction was the index that explained the spatial variability in annual $CO_2$ emissions (Nordbo et al., 2012; Velasco and Roth, 2010; Ward et al., 2015)".

*2/13 It is not correct to affirm that vegetation fraction and anthropogenic activities are correlated. Those studies analyzed the correlation between the planar fraction of green space and CO2 flux in residential neighborhoods. The biogenic component (vegetation + soil), human respiration, vehicular traffic and domestic consumption of fossil fuels have been reported as the main contributors for that particular type of urban land-use.*

We agree that term of "correlation" was invalid, and has revised by referring the statement

in Nordbo et al (2012) in Lines 79-81 as "because the green fraction has many possible factors that determine $CO_2$ emissions: a greater green fraction correlates to lesser road and population densities (Nordbo et al., 2012)".

*2/23 Define moderate urban area.*

We have added the definition of moderate urban area in this study in section for data analysis in Lines 239-240 as "Here, we defined the moderate urban area having a green fraction of 27%, which was double that of the dense urban built-up area (Table 1).".

*3/4 Table 1 does not provide information on the characteristics of the monitored sites.*

We truly apologize that Table 1 was missing and Table 2 was accidentally shown as Table 1 in the manuscript. Table 1 shows the surface cover fraction within the footprint as follow, and have been added in the revised manuscript.

Table 1. Land cover fraction within d the daytime flux footprint. Landcover classification was conducted using the Digital Map 5000 for the Kinki region in 2008 by the Geospatial Information Authority of Japan, and the green space fraction was based on a green census by the government of Sakai City. Because the land cover classification and green space are different data sources, the sum of each fraction often exceeds 100%. The daytime flux footprint was calculated using the analytical footprint model (Kormann and Meixner, 2000), and median values in 2015 were classified for sixteen direction (Fig. 1).

| | SAC west | SAC east | OPU | IZM park | IZM rural |
|---|---|---|---|---|---|
| | % | % | % | % | % |
| Residence | 27 | 9 | 9 | 1 | 15 |
| Commercial, industrial, and public office | 34 | 38 | 69 | 6 | 15 |
| Road | 27 | 29 | 10 | 3 | 6 |
| Green space | 14 | 27 | 44 | 72 | 62 |

*3/16 What does the category of others include? It is the second largest component.*

We have added the sentences for showing the details in Lines 120 -122 as "The land cover of the park consists of 51% trees, 15% grassland, and 34% other, such as ponds, buildings, pavement, and bare ground. No vehicle traffic was allowed in the park except parking.".

*3/21 Ancient tumulus?*

We have changed the word of "ancient tumulus" to "kofun (the ancient burial mound), Mozu Kofungun" in Lines 133-134. Kofun is the ancient burial mound; the tombs of kings. Around

and within the study area, there are a lot of kofun, including a kofun for Emperor Nintoku, which is the biggest tomb mound in Japan. The large keyhole-shaped kofun was shown in Fig. 1 for south of SAC and west of OPU.

*3/26-27 It means that that the instruments were mounted at 127 m above the ground, 12 times the average height of the roughness elements (i.e. buildings). Were the instruments mounted within the inertial sublayer? Fluxes obtained from very tall towers (>100 m) may not be representative of the urban landscape. Tall towers may reach above the top of the collapsing boundary layer at night, causing difficulties in interpreting the flux data.*

We believe that the measurements at SAC was within the inertial sublayer, especially in daytime. Owing to the nature of the building structure, measurements at lower height may be hampered by sporadically distributed tall buildings. We have added the sentence for the tall buildings in the revised manuscript in Lines 108-112 as "The area is a densely built-up urban area with a mean building height of $10.7 \pm 3.1$ m. Because the distributions of building heights were highly skewed toward low-height buildings, the mean building height greater than 20 m was 36 m, which occupied 33% of the total building area.".

We agree the reviewers' suggestion for uncertainties in nighttime measurement, and have added the limitation in the final paragraph in the discussion section in Lines 505-508 as "The measurement height at SAC was more than ten times higher than the mean building height, although reducing the height was restricted due to sporadic tall buildings. This could induce underestimates of nighttime fluxes (Oke, 2006), and thus, the annual emission could be underestimated.".

*3/28 Avoid grey literature.*

We have removed the grey literature.

*3/31-32 The inertial sublayer is usually located at a height of 2-4 times the mean height of the buildings. Instruments mounted at 2 m over a building of 16.2 m are clearly not placed within such layer.*

We agree that the measurement height of OPU was low, based on the fact that the mean building and tree height were 9.1 m and 13.1 m, respectively. This indicates that measured height is just 1.2 to 1.7 times the mean tree/building heights. Consequently, the measurements height were within the roughness sublayer, and must be influenced by sink/source distributions of the surface. We have added the site characteristics in the method

section in Lines 135-137 as "The mean and maximum tree heights were 13.1 ± 2.9 m and 19 m, respectively, and the mean and maximum building heights are 9.1 ± 2.9 m and 15 m, respectively.", and have added sentences for this limitation in the final paragraph in the discussion section in Lines 510-512 as "In contrast, the measurement height at OPU was within the roughness sublayer (1.2 to 1.7 times the mean building and tree heights), and thus fluxes were influenced by localized nearby fields (Oke, 2006).".

*4/11-12 The manuscript does not provide any material to support this observation.*

Thank you for pointing out the important issues. We have added the statistics in Lines 165-167 as "(RMSE = 2.18 μmol m$^{-2}$ s$^{-1}$; $F_{open}$ = 1.00 * $F_{closed}$ - 0.03 μmol m$^{-2}$ s$^{-1}$; $R^2$ = 0.84; $F_{open}$ and $F_{closed}$ represent $CO_2$ fluxes by the open and closed paths, respectively)".

*4/21 No trend removal was applied, why?*

To avoid possible underestimation of low-frequency flux contributions (Aubinet et al., 2003; Moncrieff et al., 2004), we did not apply trend removal.

*4/29 If no filtering based on u\* was applied, was the stationarity criteria used? (see Aubinet et al. Advances in Ecological Research 30, 113-175, 2000).*

As shown in an answer for the specific comment (*5/1-8*), we applied common stationary criteria (Appendix B in Ueyama et al., 2012) based on the instationary test, integral turbulent test, and high order moment (Foken and Wichura, 1996; Vickers and Mahart. 1997) for both daytime and nighttime. We checked unclear dependence of nighttime $CO_2$ flux to u\*, which was consistent with previous urban studies (e.g., Liu et al., 2012).

*4/29 Was the flux storage below the towers during night time considered?*

We considered the storage terms for IZM, but did not consider the storage term in this study because estimating spatially representative storage fluxes was difficult. We agree that mentioning the uncertainties associated with storage was important and have added the sentences in the revised manuscript in the method in Lines 206-209 section as "The storage term was added to the turbulent fluxes for the vegetative site (IZM), whereas storage was not considered for urban sites (SAC and OPU). The storage term for IZM was estimated based on $CO_2$ concentrations at the height of the eddy covariance measurements.", and have

added possible uncertainties in the final paragraph in the discussion in Lines 509-510 section as "CO$_2$ storage within the building was not considered in our study, but must be important in the late afternoon and early morning (Vogt et al., 2006).".

*5/1-8 The data quality assurance is poorly described.*

The quality assurance used in this study are based on our previous study (Appendix B in Ueyama et al., 2012). First, we removed data that did not meet stationary condition based on a method of Foken and Wichura (1996), where half-hourly data were subdivided into 6 hour, and then covariance was calculated for each 5-minute data. If difference between mean of covariance for the subdivided classes and half-hourly covariance was greater than 40% of half-hourly covariance, data was rejected as instationary. Then, we applied integral turbulent test, where turbulent intensity was greater than 50% for IZM and 200% for SAC and OPU those predicted by the similarity theory. According to the high moment test (Vickers and Mahrt. 1997), we removed data when the absolute value of skewness was greater than 3.6 or when the value of kurtosis was greater than 14.4. Those information has been added in the revised manuscript in Lines 210-225.

*5/15 The percentages of data coverage are useless if the total measurement period for each tower is not provided.*

We have added the total measurement period in Lines 240-242 as "we formed five flux datasets from measurements at the three sites in 2015 for SAC and OPU and in the period from February 2015 to January 2016 in IZM".

*5/18 CO2 sequestration or carbon capture are common terms to talk about negative fluxes. Assimilatory flux is not a common term in the community of urban climatologists.*

Since the terms "CO$_2$ sequestration" and "carbon capture" were often used for net flux instead of the gross fluxes, we have revised the term from "the assimilatory fluxes" to "gross photosynthetic flux", which could be objective.

5/18 Are there no trees at the SAC sites?

The green fraction of SAC west and east were 14% and 27%, respectively. We have added a Table 1, which was missing in the previous manuscript.

*5/19 What does the biological signal consider? Does it include the soil respiration contribution?*

We have revised the sentence to "the apparent daytime uptake was measured" in Lines 246-247.

*6/5 This reviewer does not consider such a simplistic approach can solve, even roughly, the potential role of vegetation in the urban CO2 flux. Weissert et al. Urban Climate 8, 100-125, 2014 provide a comprehensive review of methods to quantify the potential carbon sequestration by urban vegetation. Check also the approaches followed by Velasco et al. Atmos. Chem. Phys. 13, 10185-10202, 2013 and Ward et al. Atmos. Chem. Phys. 13, 4645-4666, 2013 to evaluate the urban vegetation contribution.*

We agree that our analysis for mitigation was too simplistic. Because the analysis for the mitigation did not directly relate to the conclusion of our paper, we have removed all sentences related to mitigation for this analysis.

*6/14 Biomass density is the important parameter to investigate (see Velasco et al. Landscape and Urban Planning 148, 99-107, 2016).*

Our upscaling method is a previous proposed method based on the synthesis of urban eddy covariance network at the global scale (Nordbo et al., 2012), where the green fraction was found to be a good index to explain the urban $CO_2$ fluxes at various cities. Here, we re-examined their hypothesis and similarly found that the green fraction was useful in our city. This suggests that the proposed method by Nordbo et al. (2012) could be applicable.

Since Velasco et al. (2016) compared $CO_2$ fluxes at Singapore and Mexico, and suggested that vegetation density was potentially important to explain the differences. However, climate and vegetation type among two regions are too much different, and thus the suggestion for the importance of vegetation density was not conclusive. Spatial distributions in vegetation density could not be easily available at most cities, and thus application of vegetation density is limited. We believe that the method using the green fraction, proposed by Nordbo et al. (2012), had a potential for upscaling fluxes at various cities; and thus, we first examined the method in this study. We agree that increasing number of explanatory variable might improve the scaling, and have added future improvement in discussion in Lines 480-481 as "Other environmental variables, such as biomass density (Velasco et al., 2016), might improve the scaling of $CO_2$ fluxes in various cities.".

*7/15 CO2 contributions from tombs? Do you suggest that corpses capture carbon?*

We have revised the term to "kofun", as explained above.

*7/18 Are statistically different the weekly variations? Any difference between fluxes on Saturdays and Sundays?*

Thank you for pointing out the important issue. We have conducted the F test and T test, and found that weekly variations were statistically significant in SAC east in addition to SAC west and OPU, whereas those for the IZM park and IZM rural were insignificant. We have revised the sentence in Lines 363-366 as "On average, $CO_2$ emissions on weekdays were approximately 50% greater than emissions on weekends and holidays ($p < 0.01$) at the west SAC and OPU sites, even though the weekday $CO_2$ flux at the east SAC was 10% higher than the fluxes on holidays ($p < 0.01$).", and have added the standard deviation in Figure 8. The difference between Saturdays and Sundays are insignificant, which could be shown in Figure 8.

*7/25 Table 2 was not included in the manuscript.*

We apologize that previous manuscript lost Table 1, and Table 2 was shown as Table 1. We have revised showing both Table 1 and 2.

*7/29-31 This sentence contradicts the previous statement that vegetation is correlated with anthropogenic activities (2/15).*

In previous manuscript, our intent was that mitigations by direct $CO_2$ uptake by vegetation was limited; and then, green fraction could be an index explaining the intensity of human activities rather than an index explaining direct vegetative $CO_2$ uptake. As above mentioned, in the revised manuscript, analysis of the mitigation has been removed because the analysis was weak.

*7/31 A comparison with flux data reported by Hirano et al. SOLA 11, 100-113, 2015 and Moriwaki and Kanda Journal of Applied Meteorology 43, 1700-1710, 2004 for two residential neighborhoods of Tokyo is needed.*

We have revised the discussion in Lines 404-407 as "The annual emissions rate in our urban

center was comparable to those of the densely populated residential areas in Yoyogi, Tokyo (4.3 kg C m$^{-2}$ yr$^{-1}$, Hirano et al., 2015), and Kugahara, Tokyo (3.4 kg C m$^{-2}$ yr$^{-1}$, Moriwaki and Kanda, 2004).".

*8/7-9 A proper comparison should consider differences on the urban morphology, climatology, population density, anthropogenic activities, etc.*

We have added sentences in the first paragraph in the discussion section for describing urban morphology and population density in Lines 392-398 as "$CO_2$ emission in our city was lower than those measured in urban centers: a dense urban built-up area in London (12.7 kg C m$^{-2}$ yr$^{-1}$; Ward et al., 2015), the historical city center in Florence (8.3 kg C m$^{-2}$ yr$^{-1}$; Gioli et al., 2012), and a residential area of south central Vancouver (6.7 kg C m$^{-2}$ yr$^{-1}$; Christen et al., 2012). The annual emissions in our city were also lower than previous cities that had a similar population density; there were only two cities whose populations were higher than that in our city, but the annual emissions in our city were seventh in the global synthesis (Fig. 12b in Ward et al., 2015).".

*8/11 Be consistent with the use of units.*

Thank you for mentioning the units; we have revised the unit by using "C" instead of "$CO_2$" for the annual balance.

*8/20 Such slopes cannot be appreciated in Fig. 4.*

We have recheck the statistics and found that the slope for SAC west was insignificant. We have revised the sentence in Lines 413-416 as "These values are comparable to those obtained in our city: -0.37 g C m$^{-2}$ d$^{-1}$ °C$^{-1}$ for all SAC ($p = 0.03$) and -0.27 g C m$^{-2}$ d$^{-1}$ °C$^{-1}$ for east SAC ($p < 0.01$), when mean air temperatures were less than 15°C (Fig. 7), although the correlation for west SAC was insignificant.".

*8/20-21 The green dots in Fig 4(b) show a clear variation with temperature, at least in the 10-35 degC range.*

Thank you for mentioning this. We have added the sentence describing the rate in the result section in Lines 349-352 as "In the urban park and rural area, $CO_2$ emissions decreased as temperatures increased above 15°C: -0.27 g C m$^{-2}$ d$^{-1}$ °C$^{-1}$ for the urban park ($p < 0.01$) and

-0.13 g C m$^{-2}$ d$^{-1}$ °C $^{-1}$ for the rural area ($p < 0.01$) when the mean air temperatures were greater than 15°C (Fig. 7)".

*9/10 Migration of urban emissions to urban parks?*

We have revised the sentence in Line 345 as "The urban park acted as a net annual CO$_2$ source despite the abundant vegetation.".

*9/20-21 Indeed, a comprehensive partitioning of the CO2 flux as a function of the upcoming wind direction may be useful to understand differences according to the surface characteristics. However, such partitioning may suffer of severe uncertainties due to advection issues triggered by wind shifts during the averaging periods.*

We agree that the methods had uncertainties associated with advection, and have added a sentence in the final paragraph in the discussion section showing our limitations in Lines 513-514 as "Separating wind sectors using the footprint analysis may suffer uncertainties when advection was trigged by wind shifts.".

*9/29 Green fraction as an index of human activities?*

We have revised the sentence as the reviewers' suggestion in Lines 476-477 as "the green fraction was an index of human activities (Nordbo et al., 2012)".

*10/2-3 This is a reason why eddy covariance flux towers cannot solve by themselves the puzzle of the greenhouse gas emissions in urban ecosystems.*

We agree that it is difficult to evaluate gross emission using the eddy covariance (EC) data, but we believe that the comparison among different methods is important. Each estimate has advantage and disadvantage. For example, EC can estimate net fluxes including emissions from the flux footprint and vegetative uptake, and they do not include point source emissions, such as power plant located outside a city. We believe that constructing the flux map based on the EC data is the important first step for understanding CO$_2$ fluxes in urban landscape.

Currently, we do not want to estimate total emissions in this study. In this study, we would like to present the applicability of the previously proposed upscaling method for the global scale (Nordbo et al., 2012) for our single city. The upscaling EC data could be useful for other research fields, such as regional atmospheric inversion if we can added the point source

emissions.

We agree that such upscaling suffered considerable uncertainties, and thus we have added further discussion for limitation in the revised manuscript in Lines 500-503 as "Because our simple method potentially contained uncertainties associated with a limited number of one-year eddy covariance sites, and only the consideration of the green fraction, the estimates should be improved with further eddy covariance sites and additional environmental variables in order to explain $CO_2$ fluxes.".

*10/3-4 Such comparison should consider the extrapolated flux after subtracting the "biological" component.*

As described above answer, we intended to show differences among the two different methods, rather than to estimate gross emissions by the upscaling. It is valid to know how the magnitudes differ among two estimates (inventory and upscaling eddy covariance data): inventory-based estimates considered point source emissions that was located outside the city, but did not consider the biological uptake. The comparison could be used for other research fields.

References:

Aubinet, M., Clement, R. C., Elbers, J. A., Foken, T., Grelle, A.and co-authors. 2003. Methodology for data acquisition, storage, and treatment. In: Fluxes of Carbon, Water and Energy of European Forests (ed. R. Valentini), Springer-Verlag, Berlin, pp. 9-35.

Högström, U., Bergstoröm, H., Alexandersson, H., 1982. Turbulence characteristics in a near neutrally stratified urban atmosphere. Boundary Layer Meteorol. 23, 449-472.

Järvi, L., Nordbo, A., Riikonen, A., Moilanen, J., Nikinmaa, E., Vesala, T., 2012. Seasonal and annual variation of carbon dioxide surface fluxes in Helsinki, Finland, in 2006-2010. Atmos. Chem. Phys., 12, 8475-8489.

Kaimal, J.C., Finnigan, J.J., 1994. Atmospheric Boundary Layer Flows, 289pp., Oxford Univ. Press, New York.

Liu, H.Z., Feng, J.W., Vesala, T., 2012. Four-year (2006-2009) eddy covariance measurements of $CO_2$ flux over an urban area in Beijing. Atmos. Chem. Phys., 12, 7881-7892.

Moncrieff, J., Clement, R., Finnigan, J. and Meyers, T. 2004. Averaging, detrending, and filtering of eddy covariance time series. In: Handbook of Micrometeorology: A Guide for Surface Flux Measurement and Analysis (eds. X. Lee, W. Massman and B. Law), Kluwer Academic Publishers, Dordrecht, pp. 7-31.

Nordbo, A., Järvi, L., Haapanala, S., Wood, C. R., Vesala, T., 2012. Fraction of natural area as main

predictor of net $CO_2$ emissions from cities. Goephys. Res. Lett. 39, doi:10.1029/2012GL053087.

Ueyama, M., Hirata, R., Mano, M., Hamotani, K., Harazono, Y., Hirano, T., Miyata, A., Takagi, K., Takahashi, Y., 2012. Influences of various calculation options on heat, water and carbon fluxes determined by open- and closed-path eddy covariance methods. Tellus 64B, 19048, doi.org/10.3402/tellusb.v64i0.19048.

Ueyama, M., Takanashi, S., Takahashi, Y., 2014. Inferring methane fluxes at a larch forest using Lagrangian, Eulerian, and hybrid inverse models. J. Geophys., Res. Biogeosciences, doi.10.1002/2014JG002716.

Velasco, E., Roth, M., 2010. Cities as net source of $CO_2$: review of atmospheric $CO_2$ exchange in urban environments measured by eddy covariance technique. Geography Compass 4/9, 1238-1259.

Ward, H.C., Kotthaus, S., Grimmond, C.S.B., Bjorkegren, A., Wilkinson, M., Morrison, W.T.J., Evans, J.G., Morison, J.I.L., Iamarino, M. 2015. Effects of urban density on carbon dioxide exchanges: observations of dense urban, suburban and woodland areas of southern England. Environ. Pollut. 198, 189-200.

Wilson, K., Goldstein, A., Falge, E., Aubinet, M., Baldocchi, D. and co-authors. 2002. Energy balance closure at FLUXNET sites. Agric. Forest Meteorol. 113, 223-243.

**Responses to reviewer's comments 2.**

Ueyama and Ando
Diurnal, weekly, seasonal and spatial variabilities in carbon dioxide flux in different urban landscapes in Sakai, Japan

*General comment: This study presents flux of CO2 measured at three different types of locations (urban suburb and rural) in Sakai, Osaka. The fluxes have been calculated using the eddy covariance method. In summary, the paper presents important result about the variations of flux in different time scales. The data looks reliable and worth reporting. However, I am not fully satisfied with the discussion which is largely qualitative and very brief. Therefore, there is a significant scope to improve the draft considering following aspects.*

Thank you very much for your constructive comments, editing, and encouragement. Based on the reviewers' comments, we have revised the manuscript. The revised manuscript has been edited using a NPG Language editing service. All revisions for reviewer 2 are marked as green and revisions for reviewer 1 are marked as red, in the revised manuscript. The edited by the NPG Language editing service was marked as purple.

*(1) The experimental uncertainties in the measurements of eddy parameters (CO2, winds, etc.) have not been reported. In the "Observations" section, the errors and calibration procedure should be presented in details. The variations of meteorological parameters (wind parameters, RH, temp) should be presented (Figures, wind rose) for each season.*

We have added the description about the intercomparison about $CO_2$ fluxes from the open- and closed-path eddy covariance for the OPU in Lines 165-166, calibration information in Lines 175-183, flow statistics for validity of turbulence in Lines 212-217 for showing measurement uncertainties.

We agree that showing meteorology is very important. We have also added the new section "*3-1. Meteorological characteristics*" in Lines 287-300 and new figures (Fig. 2 & 3) for showing general characteristics including air temperature, humidity (VPD), precipitation, and wind rose. Wind rose in each season has also been added in Fig. A1.

*(2) The authors should work to make proper statistical representations of results (using mean, median, percentiles, standard deviation, etc.). None of the representations (figures) used in the paper show the variance of CO2 flux on daily, weekly and monthly scales. Accordingly, the discussion is "overall"*

*but not the "detailed".*

We have added the statistic throughout the documents. For example, in the section "*3-2. Diurnal variations*", statistical analyses and information about statistical significance have been added in Lines 303-329. We have added standard deviations in Fig. 8 and statistics in Lines 350-352, and Lines 364-365 for representing variation in the weekly analysis. Including variances in the all figures are difficult, because large variances associated with other factors may mask the mean values. Consequently, we have added new Fig. 5 to show how half-hourly data had variance.

*(3) About the results and discussion, sometimes I am confused to see overlaps between the interpretations of data for different sites. Hence, the important governing processes at each site. This is because the discussion is not structured and very brief. For example, the results of "3.1 Diurnal variations" "3.3 Weekly variations" at all sites have been summarized just in few lines. Therefore, it is difficult to follow and appreciate the discussion.*

We have fully revised the section for "*3.2 Diurnal variations*" for adding further details in half-hourly fluxes in Lines 302-329, where new analysis for light-dependency of $CO_2$ fluxes in IZM and OPU in Lines 266-272 and new Figure 5, showing important governing processes. Consequently, we have divided this section into two paragraphs for showing fluxes in urban built-up and rural/park areas. Since statement in "*3.4 Weekly variations*" could be simple, we have only added statistics for showing significance.

*(4) Additional analysis: It would be interesting to see how the diurnal flux changes with weather conditions in each season. For this, I suggest to separate the data, at least for rainy, cloudy and clear-sky days in each season. Please provide the diurnal flux figures measured under distinct weather conditions (rainy, cloudy and clear sky)*

We agree that it is interesting to show the diurnal fluxes with different weather conditions. We have added this in new Figure A2, where the daytime $CO_2$ emissions for the urban built-up was large in sunny days (Fig. A2a, b), and daytime uptake in the rural area was greater in sunny days (Fig. A2e). We have added these results in "*3.2 Diurnal variations*" in Lines 210-314 and Lines 323-329.

*(5) English should be improved, sometimes choice of word and phrase are not appropriate.*

We apologize for our poor English. Although the previous manuscript was edited using the NPG Language editing service, we have edited again the revised manuscript using the NPG Language editing service. The edited sentences have been shown as purple.

*(6) Overall, the paper looks a kind of well written report. However, a scientific paper requires more detailed representations of both results and discussion. Some specific comments are given here:*

Thank you for your constructive comments. We have revised the manuscript based on your general and specific comments.

*Page 1 Line 14-16: Following sentence is ambiguous and needs to be re-written. "In contrast, the dense and moderately urban areas exhibited higher emissions in winter and summer months, when emissions significantly increased as air temperature increased in summer and air temperature decreased in winter."*

We have revised the sentence as "In contrast, the dense and moderately urban areas emitted $CO_2$ in all seasons. $CO_2$ emissions in the urban areas were high in the winter and summer months, and they significantly increased with the increase in air temperature in the summer and the decrease in air temperature in the winter." in Lines 29-33.

*Page 1 Line 25: In this sentence, I do not find the logic to use "Consequently, ..."*

We have revised the sentence as "Urban areas account for only a small percentage of the earth's land surface but emit 30–50% of total anthropogenic $CO_2$ (Mills, 2007; Stterthwaite, 2008), and thus, cities are important sources of the global $CO_2$ emissions. The $CO_2$ emissions among global cities are highly heterogeneous (Mills, 2007; Nordbo et al., 2012), and the temporal variability is high (Velasco and Roth, 2010). To evaluate the spatio-temporal variabilities in $CO_2$ emissions for global cities, studies using multiple methods, such as measurements (Velasco and Roth, 2010) and emission inventories (Oda and Maksyutov, 2011), are currently conducted." in Lines 45-52.

*Page 1 Line 27: " Global CO2 emissions have often been estimated using inventories ..." I do not understand this, what do you mean by "inventories" here ?*

We have revised the sentence as "Global $CO_2$ emissions have often been estimated using emission inventories based on point source databases, statistics for national and regional

$CO_2$ emissions, and satellite remote sensing (Oda and Maksyutov, 2011)." in Lines 53-55.

*Page 2 Line 15: "because vegetation fraction can be correlated with anthropogenic activities. " How?, an explanation is required.*

We have revised the sentence as "Multi-site eddy covariance towers were used to synthesize the data and showed that green fraction was the index that explained the spatial variability in annual $CO_2$ emissions (Nordbo et al., 2012; Velasco and Roth, 2010; Ward et al., 2015), because the green fraction has many possible factors that determine $CO_2$ emissions: a greater green fraction correlates to lesser road and population densities (Nordbo et al., 2012)." in Lines 77-81.

*Page 3 line 1: " according to the apanese Meteorological " Spelling issue, please correct*

We have corrected the typo.

*Page 4 line 19: Following sentence needs correction "using" has come twice Turbulent fluxes were calculated using the eddy covariance method using the Flux Calculator program*

We have revised the sentence as "Turbulent fluxes were calculated with the eddy covariance method using the Flux Calculator program (Ueyama et al., 2012)." in Lines 187-189.

*Page 6 " 3.1 Diurnal variations" The discussion in this section is very qualitative. It is needed to be more quantitative in terms of site to site variations represented by suitable diurnal statistical analysis?*

As answered in general comments 2, 3, and 4, the section for "*3.2 Diurnal variations*" has been fully revised shown in Lines 303-329.

*Page 6 3.2 Seasonal variations Again, the discussion in this section is very qualitative. It is needed to be more quantitative in terms of site to site variations represented by suitable seasonal statistical analysis?*

Description for seasonal variations, we has added statistics in Lines 346, and 350-352. Further discussion for the seasonal variations has been added in the discussion section in Lines 414-415, and 427-436, based on the reviewer 1 comments.

*Page 6 Line 22-23, Previous studies ...............our city. This sentence is not clear, what authors wish to convey?*

We have revised the sentence as "The green fraction was a useful index that explained the spatial variability in the annual $CO_2$ fluxes, as suggested in global scale studies (Nordbo et al., 2012; Velasco and Roth, 2010). The relationship based on eddy covariance data within a single city could be useful to evaluate $CO_2$ emissions at the city scale." in Lines 526-530.